# WHAT THEY DO WHEN IN DOUBT: A STUDY OF INDUCTIVE BIASES IN SEQ2SEQ LEARNERS

**Eugene Kharitonov**[*]
Facebook AI

**Rahma Chaabouni**[*]
Facebook AI / ENS Ulm

## ABSTRACT

Sequence-to-sequence (seq2seq) learners are widely used, but we still have only limited knowledge about what inductive biases shape the way they generalize. We address that by investigating how popular seq2seq learners generalize in tasks that have high ambiguity in the training data. We use four new tasks to study learners' preferences for memorization, arithmetic, hierarchical, and compositional reasoning. Further, we connect to Solomonoff's theory of induction and propose to use description length as a principled and sensitive measure of inductive biases.

In our experimental study, we find that LSTM-based learners can learn to perform counting, addition, and multiplication by a constant from a *single training example*. Furthermore, Transformer and LSTM-based learners show a bias toward the hierarchical induction over the linear one, while CNN-based learners prefer the opposite. The latter also show a bias toward a compositional generalization over memorization. Finally, across all our experiments, description length proved to be a sensitive measure of inductive biases.

## 1 INTRODUCTION

Sequence-to-sequence (seq2seq) learners (Sutskever et al., 2014) demonstrated remarkable performance in machine translation, story generation, and open-domain dialog (Sutskever et al., 2014; Fan et al., 2018; Adiwardana et al., 2020). Yet, these models have been criticized for requiring a tremendous amount of data and being unable to generalize systematically (Dupoux, 2018; Loula et al., 2018; Lake & Baroni, 2017; Bastings et al., 2018). In contrast, humans rely on their inductive biases to generalize from a limited amount of data (Chomsky, 1965; Lake et al., 2019). Due to the centrality of humans' biases in language learning, several works have studied inductive biases of seq2seq models and connected their poor generalization to the lack of the "right" biases (Lake & Baroni, 2017; Lake et al., 2019).

In this work, we focus on studying inductive biases of seq2seq models. We start from an observation that, generally, multiple explanations can be consistent with a limited training set, each leading to different predictions on unseen data. A learner might prefer one type of explanations over another in a systematic way, as a result of its *inductive biases* (Ritter et al., 2017; Feinman & Lake, 2018).

To illustrate the setup we work in, consider a quiz-like question: if $f(3)$ maps to 6, what does $f(4)$ map to? The "training" example is consistent with the following answers: 6 ($f(x) \equiv 6$); 7 ($f(x) = x + 3$); 8 ($f(x) = 2 \cdot x$); any number $z$, since we always can construct a function such that $f(3) = 6$ and $f(4) = z$. By analyzing the learner's output on this new input, we can infer its biases.

This example demonstrates how biases of learners are studied through the lenses of the poverty of the stimulus principle (Chomsky, 1965; 1980): if nothing in the training data indicates that a learner should generalize in a certain way, but it does nonetheless, then this is due to the biases of the learner. Inspired by the work of Zhang et al. (2019) in the image domain, we take this principle to the extreme and study biases of seq2seq learners in the regime of very few training examples, often as little as one. Under this setup, we propose four new synthetic tasks that probe seq2seq learners' preferences to memorization-, arithmetic-, hierarchical- and compositional-based "reasoning".

---

[*]Equal contribution.

Next, we connect to the ideas of Solomonoff's theory of induction (Solomonoff, 1964) and Minimal Description Length (Rissanen, 1978; Grunwald, 2004) and propose to use description length, under a learner, as a principled measure of its inductive biases.

Our experimental study[1] shows that the standard seq2seq learners have strikingly different inductive biases. We find that LSTM-based learners can learn non-trivial counting-, multiplication-, and addition-based rules from as little as one example. CNN-based seq2seq learners would prefer linear over hierarchical generalizations, while LSTM-based ones and Transformers would do just the opposite. When investigating the compositional reasoning, description length proved to be a sensitive measure. Equipped with it, we found that CNN-, and, to a lesser degree, LSTM-based learners prefer compositional generalization over memorization when provided with enough composite examples. In turn, Transformers show a strong bias toward memorization.

## 2 Searching for inductive biases

To formalize the way we look for inductive biases of a learner $\mathcal{M}$, we consider a training dataset of input/output pairs, $T = \{x_i, y_i\}_{i=1}^n$, and a hold-out set of inputs, $H = \{x_i\}_{i=n+1}^k$. W.l.o.g, we assume that there are two candidate "rules" that explain the training data, but do not coincide on the hold-out data: $\mathcal{C}_1(x_i) = \mathcal{C}_2(x_i) = y_i, \ 1 \leq i \leq n$ and $\exists i : \mathcal{C}_1(x_i) \neq \mathcal{C}_2(x_i), n + 1 \leq i \leq k$.

To compare preferences of a learner $\mathcal{M}$ toward those two rules, we fit the learner on the training data $T$ and then compare its predictions on the hold-out data $H$ to the outputs of the rules. We refer to this approach as "intuitive". Usually, the measures of similarity between the outputs are task-specific: McCoy et al. (2020) used accuracy of the first term, Zhang et al. (2019) used correlation and MSE, and Lake & Baroni (2017) used accuracy calculated on the entire output sequence.

We too start with an accuracy-based measure. We define the fraction of perfect agreement (FPA) between a learner $\mathcal{M}$ and a candidate generalization rule $\mathcal{C}$ as the fraction of seeds that generalize *perfectly* in agreement with that rule on the hold-out set $H$. Larger FPA of $\mathcal{M}$ is w.r.t. $\mathcal{C}$, more biased $\mathcal{M}$ is toward $\mathcal{C}$. However, FPA does not account for imperfect generalization nor allows direct comparison between two candidate rules when both are dominated by a third candidate rule. Hence, below we propose a principled approach based on the description length.

**Description Length and Inductive Biases** At the core of the theory of induction (Solomonoff, 1964) is the question of continuation of a finite string that is very similar to our setup. Indeed, we can easily re-formulate our motivating example as a string continuation problem: "$3 \rightarrow 6; 4 \rightarrow$". The solution proposed by Solomonoff (1964) is to select the continuation that admits "the simplest explanation" of the entire string, i.e. that is produced by programs of the shortest length (description length).

Our intuition is that when a continuation is "simple" for a learner, then this learner is biased toward it. We consider a learner $\mathcal{M}$ to be biased toward $\mathcal{C}_1$ over $\mathcal{C}_2$ if the training set and its extension according to $\mathcal{C}_1$ has a shorter description length (for $\mathcal{M}$) compared to that of $\mathcal{C}_2$. Denoting description length of a dataset $D$ under the learner $\mathcal{M}$ as $L_{\mathcal{M}}(D)$, we hypothesise that if $L_{\mathcal{M}}(\{\mathcal{C}_1(x_i)\}_{i=1}^k) < L_{\mathcal{M}}(\{\mathcal{C}_2(x_i)\}_{i=1}^k)$, then $\mathcal{M}$ is biased toward $\mathcal{C}_1$.

**Calculating Description Length** To find the description length of data under a fixed learner, we use the online (prequential) code (Rissanen, 1978; Grunwald, 2004; Blier & Ollivier, 2018).

The problem of calculating $L_{\mathcal{M}}(D)$, $D = \{x_i, y_i\}_{i=1}^k$ is considered as a problem of transferring outputs $y_i$ one-by-one, in a compressed form, between two parties, Alice (sender) and Bob (receiver). Alice has the entire dataset $\{x_i, y_i\}$, while Bob only has inputs $\{x_i\}$. Before the transmission starts, both parties agreed on the initialization of the model $\mathcal{M}$, order of the inputs $\{x\}$, random seeds, and the details of the learning procedure. Outputs $\{y_i\}$ are sequences of tokens from a vocabulary $V$. W.l.o.g. we fix some order over $\{x\}$. We assume that, given $x$, the learner $\mathcal{M}$ produces a probability distribution over the space of the outputs $y$, $p_{\mathcal{M}}(y|x)$.

---

[1]Code used in the experiments can be found at `https://github.com/facebookresearch/FIND`.

The very first output $y_1$ can be sent by using not more than $c = |y_1| \cdot \log |V|$ nats, using a naïve encoding.[2] After that, both Alice and Bob update their learners using the example $(x_1, y_1)$, available to both of them, and get identical instances of $\mathcal{M}_1$.

Further transfer is done iteratively under the invariant that both Alice and Bob start every step $t$ with exactly the same learners $\mathcal{M}_{t-1}$ and finish with identical $\mathcal{M}_t$. At step $t$ Alice would use $\mathcal{M}_{t-1}$ to encode the next output $y_t$. This can be done using $\left( - \log p_{\mathcal{M}_{t-1}}(y_t|x_t) \right)$ nats (MacKay, 2003). Since Bob has exactly the same model, he can decode the message to obtain $y_t$ and use the new pair $(x_t, y_t)$ to update his model and get $M_t$. Alice also updates her model, and proceeds to sending the next $y_{t+1}$ (if any), encoding it with the help of $\mathcal{M}_t$. The cumulative number of nats transmitted is:

$$L_{\mathcal{M}}(D) = -\sum_{t=2}^{k} \log p_{\mathcal{M}_{t-1}}(y_t|x_t) + c. \tag{1}$$

The obtained code length of Eq. 1 depends on the order in which $y$ are transmitted and the procedure we use to update $\mathcal{M}$. To account for that, we average out the data order by training with multiple random seeds. Further, for larger datasets, full re-training after adding a new example is impractical and, in such cases, examples can be transmitted in blocks.

If we measure the description length of the training data $T$ shuffled with the hold-out data $H$, both datasets would have symmetric roles. However, there is a certain asymmetry in the extrapolation problem: we are looking for an extrapolation from $T$, not vice-versa. To break this symmetry, we always transmit outputs for the entire training data as the first block.

While $L_{\mathcal{M}}(D)$ is seemingly different from the "intuitive" measures introduced before, we can illustrate their connection as follows. Consider a case where we first transmit the training outputs as the first block and *all* of the hold-out data outputs under $\mathcal{C}, \mathcal{C}(H)$, as the second block. Then the description length is equal to cross-entropy of the trained learner on the hold-out data, recovering a process akin to the "intuitive" measuring of inductive biases. With smaller blocks, the description length also catches whether a learner is capable of finding regularity in the data fast, with few data points; hence it also encodes the speed-of-learning ideas for measuring inductive biases (Chaabouni et al., 2019).

Finally, the description length has three more attractive properties when measuring inductive biases: (a) it is domain-independent, i.e. can be applied for instance in the image domain, (b) it allows comparisons across models that account for model complexity, and (c) it enables direct comparison between two candidate rules (as we will show in the Section 5).

## 3 TASKS

We describe four tasks that we use to study inductive biases of seq2seq learners. We select those tasks to cover different aspects of learners' behavior. Each task has a highly ambiguous training set which is consistent with infinite number of generalizations. We pre-select several candidate rules highlighting biases that are useful for language processing and known to exist in humans, or are otherwise reasonable. Our experiments show that these rules cover many cases of the learners' behavior.

The first two tasks study biases in arithmetic reasoning: *Count-or-Memorization* quantifies learners' preference for counting vs. a simple memorization and *Add-or-Multiply* further probes the learners' preferences in arithmetic operations. We believe these tasks are interesting, as counting is needed in some NLP problems like processing linearized parse trees (Weiss et al., 2018). The third task, *Hierarchical-or-Linear*, contrasts hierarchical and linear inductive reasoning. The hierarchical reasoning bias is believed to be fundamental in learning some syntactical rules in human acquisition of syntax (Chomsky, 1965; 1980). Finally, with the *Composition-or-Memorization* task, we investigate biases for systematic compositionality, which are central for human capabilities in language. Figure 1 illustrates these four tasks.

---

[2]As we are interested in comparing candidate generalization rules, the value of the additive constant $c$ is not important, as it is learner- and candidate-independent. In experiments, we subtract it from all measurements.

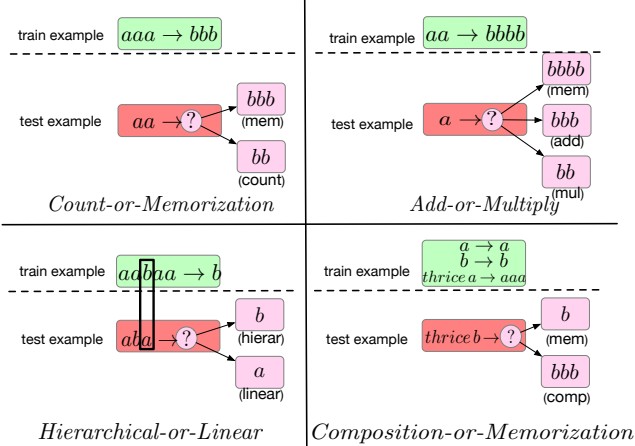

Figure 1: Illustration of the tasks. After training on the train examples (green blocks), learners are tested on held-out examples (red blocks). In pink blocks are generalizations according to the candidate rules.

By $a^k$ we denote a sequence that contains token $a$ repeated $k$ times. For training, we represent sequences in a standard way: the tokens are one-hot-encoded separately, and we append a special end-of-sequence token to each sequence. Input and output vocabularies are disjoint.

**Count-or-Memorization**: In this task, we contrast learners' preferences for counting vs. memorization. We train models to fit a single training example with input $a^l$ and output $b^l$ (i.e., to perform the mapping $a^l \rightarrow b^l$) and test it on $a^m$ with $m \in [l-10, l+10]$. If a learner learns the constant function, outputting $b^l$ independently of its inputs, then it follows the mem strategy. On the other hand, if it generalizes to the $a^m \rightarrow b^m$ mapping, then the learner is biased toward the count strategy.

**Add-or-Multiply**: This task is akin to the motivating example in Section 1. The single training example represents a mapping of an input string $a^l$ to an output string $b^{2l}$. As test inputs, we generate $a^m$ for $m$ in the interval $[l-3, l+3]$. We consider the learned rule to be consistent with mul if for all $m$, the input/output pairs are consistent with $a^m \rightarrow b^{2m}$. Similarly, if they are consistent with $a^m \rightarrow b^{m+l}$, we say that the learner follows the addition rule, add. Finally, the learner can learn a constant mapping $a^m \rightarrow b^{2l}$ for any $m$. Again, we call this rule mem.

**Hierarchical-or-Linear**: For a fixed depth $d$, we train learners on four training examples $x^d y x^d \rightarrow y$ where $x, y \in \{a, b\}$.[3] Each training example has a nested structure, where $d$ defines its depth. A learner with a hierarchical bias (hierar), would output the middle symbol. We also consider the linear rule (linear) in which the learner outputs the $(d+1)^{th}$ symbol of its input.
To probe learners' biases, we test them on inputs with different depths $m \in [d-2, d+2]$. Note that to examine the linear rule (i.e. if the learner outputs the $(d+1)^{th}$ symbol of any test input of depth $m$), we need $m \geq \frac{d}{2}$. Similar to the previous tasks, there is no vocabulary sharing between a model's inputs and outputs (input and output tokens $a$ and $b$ are different).

**Composition-or-Memorization**: We take inspiration from SCAN (Lake & Baroni, 2017), a benchmark used for studying systematic generalization of seq2seq learners.[4] The input vocabulary has $N$ symbols $a_i$ that are one-to-one mapped into $N$ output symbols $b_i$ (i.e., $a_i \rightarrow b_i$). In addition, there is a modifier token $thrice$: when $thrice$ precedes an input symbol $a_i$, the corresponding output is repeated three times: $thrice\ a_i \rightarrow b_i b_i b_i$.

We train a learner on all non-compositional examples ($a_i \rightarrow b_i$) and $M$ ($M < N$) compositional examples ($thrice\ a_i \rightarrow b_i b_i b_i$). At test time, we feed the learner with the remaining compositional examples ($thrice\ a_i, i > M$). If the learner generalizes to the mapping $thrice\ a_i \rightarrow b_i b_i b_i$ for $i > M$, we consider it to be biased toward a compositional reasoning (comp). As an alternative generalization,

---

[3]This mapping consist then on four combinations $a^d b a^d \rightarrow b$; $a^d a a^d \rightarrow a$; $b^d a b^d \rightarrow a$ and $b^d b b^d \rightarrow b$.

[4]In Appendix, we report a study of inductive biases on the SCAN data.

we consider a mapping where all inputs containing $a_i$ are mapped into $b_i$: $thrice \; a_i \rightarrow b_i \; (i > M)$. We call this generalization memorization (mem).

## 4 METHODOLOGY

### 4.1 SEQUENCE-TO-SEQUENCE LEARNERS

We experiment with three standard seq2seq models: LSTM-based seq2seq (LSTM-s2s) (Sutskever et al., 2014), CNN-based seq2seq (CNN-s2s) (Gehring et al., 2017), and Transformer (Vaswani et al., 2017). All share a similar Encoder/Decoder architecture (Sutskever et al., 2014).

**LSTM-s2s** Both Encoder and Decoder are implemented as LSTM cells (Hochreiter & Schmidhuber, 1997). Encoder encodes its inputs incrementally from left to right. We experiment with architectures without (LSTM-s2s no att.) and with (LSTM-s2s att.) an attention mechanism (Bahdanau et al., 2014). For the first three tasks, both Encoder and Decoder are single-layer LSTM cells with hidden size of 512 and embedding of dimension 16.

**CNN-s2s** Encoder and Decoder are convolutional networks (LeCun et al., 1990), followed by GLU non-linearities (Dauphin et al., 2017) and an attention layer. To represent positions of input tokens, CNN-s2s uses learned positional embeddings. Encoder and Decoder networks have one layer with 512 filters and a kernel width of 3. We set the embedding size to 16.

**Transformer** Encoder and Decoder are implemented as a sequence of (self-)attention and feed-forward layers. We use sinusoidal position embeddings. Both Encoder and Decoder contain one transformer layer. The attention modules have 8 heads, feed-forward layers have dimension of 512 and the embedding is of dimension 16.

In Appendix we report experiments where we vary hyperparameters of the learners.

### 4.2 TRAINING AND EVALUATION

For all tasks, we follow the same training procedure. We train with Adam optimizer (Kingma & Ba, 2014) for 3000 epochs. The learning rate starts at $10^{-5}$ and increases for the first 1000 warm-up updates till reaching $10^{-3}$. We include all available examples in a single batch. We use teacher forcing (Goodfellow et al., 2016) and set the dropout probability to 0.5. For each learner, we perform training and evaluation 100 times, changing random seeds. At generation, to calculate FPA, we select the next token greedily. We use the model implementations from fairseq (Ott et al., 2019).

As discussed in Section 2, when calculating $L$, we use the training examples as the first transmitted block at $t = 1$. In *Count-or-Memorization* and *Add-or-Multiply* this block contains one example, in *Hierarchical-or-Linear* it has 4 examples, and in *Composition-or-Memorization* it has $N + M$ examples. Next, we transmit examples obtained from the candidate rules in a randomized order, by blocks of size 1, 1, 1, and 4 for *Count-or-Memorization*, *Add-or-Multiply*, *Composition-or-Memorization*, and *Hierarchical-or-Linear* respectively. At each step, the learner is re-trained from the same initialization, using the procedure and hyper-parameters as discussed above.

Our training setup is typical for seq2seq training. It does not include any additional pressure towards description length minimization. Description length is solely used as a measure of inductive biases.

## 5 EXPERIMENTS

**Count-or-Memorization** We investigate here learners' biases toward count and mem rules. We provide a single example $a^l \rightarrow b^l$ as the training set, varying $l \in \{10, 20, 30, 40\}$. We report the learners' performances in Table 1a. We observe that, independently of the length of the training example $l$, CNN-s2s and Transformer learners inferred perfectly the mem rule with FPA-mem > 0.90 (i.e. more than 90% of the random seeds output $b^l$ for any given input $a^m$).

However, LSTM-based learners demonstrate a more complex behavior. With $l = 10$, both learners (with and without attention) exhibit a preference for mem. Indeed, while these learners rarely generalize perfectly to any of the hypothesis (0.0 FPA (no att.), 0.2/0.0 FPA for mem/count (att.)), they have significantly lower $L$-mem. As $l$ increases, LSTM-based learners become more biased

| | length $l$ | FPA ↑ | | $L$, nats ↓ | |
|---|---|---|---|---|---|
| | | count | mem | count | mem |
| LSTM-s2s no att. | 40 | 1.00 | 0.00 | **0.01**$^*$ | 97.51 |
| | 30 | 0.97 | 0.00 | **0.01**$^*$ | 72.67 |
| | 20 | 0.07 | 0.00 | **2.49**$^*$ | 55.67 |
| | 10 | 0.00 | 0.00 | 88.27 | **48.67**$^*$ |
| LSTM-s2s att. | 40 | 0.99 | 0.00 | **7.84**$^*$ | 121.48 |
| | 30 | 0.96 | 0.02 | **1.14**$^*$ | 83.48 |
| | 20 | 0.70 | 0.16 | **5.73**$^*$ | 49.33 |
| | 10 | 0.00 | 0.20 | 98.12 | **8.46**$^*$ |
| CNN-s2s | $\{10, 20, 30, 40\}$ | 0.00 | $> 0.90$ | $> 592.92$ | **<1.31**$^*$ |
| Transformer | $\{10, 20, 30, 40\}$ | 0.00 | $> 0.97$ | $> 113.30$ | **<11.14**$^*$ |

(a) Count-or-Memorization

| | length $l$ | FPA ↑ | | | $L$, nats ↓ | | |
|---|---|---|---|---|---|---|---|
| | | add | mul | mem | add | mul | mem |
| LSTM-s2s no att. | 20 | 0.00 | 0.94 | 0.00 | 25.42 | **0.31**$^*$ | 57.32 |
| | 15 | 0.07 | 0.65 | 0.00 | 19.24 | **4.67**$^*$ | 43.65 |
| | 10 | 0.95 | 0.01 | 0.00 | **0.68**$^*$ | 26.58 | 25.15 |
| | 5 | 0.04 | 0.00 | 0.00 | **17.12** | 50.83 | 18.60 |
| LSTM-s2s att. | 20 | 0.00 | 0.98 | 0.00 | 30.26 | **1.40**$^*$ | 58.84 |
| | 15 | 0.15 | 0.83 | 0.00 | 20.18 | **4.07**$^*$ | 46.36 |
| | 10 | 0.40 | 0.28 | 0.18 | **13.69** | 18.16 | 26.44 |
| | 5 | 0.00 | 0.00 | 0.97 | 45.88 | 77.86 | **0.01**$^*$ |
| CNN-s2s | $\{5, 10, 15, 20\}$ | 0.00 | 0.00 | 1.0 | $> 318.12$ | $> 346.19$ | **0.00**$^*$ |
| Transformer | $\{5, 10, 15, 20\}$ | 0.00 | 0.00 | 1.0 | $> 38.77$ | $> 50.64$ | **<3.50**$^*$ |

(b) Add-or-Multiply

| | FPA ↑ | | $L$, nats ↓ | |
|---|---|---|---|---|
| | hierar | linear | hierar | linear |
| LSTM-s2s no att. | 0.05 | 0.00 | **31.04**$^*$ | 61.84 |
| LSTM-s2s att. | 0.30 | 0.00 | **26.32**$^*$ | 57.2 |
| CNN-s2s | 0.00 | 1.00 | 202.64 | **0.00**$^*$ |
| Transformer | 0.69 | 0.00 | **4.84**$^*$ | 35.04 |

(c) Hierarchical-or-Linear with depth $d = 4$

| | $M$, examples | FPA ↑ | | $L$, nats ↓ | |
|---|---|---|---|---|---|
| | | comp | mem | comp | mem |
| LSTM-s2s no att. | 36 | 0.00 | 0.00 | 42.65 | **38.55** |
| | 24 | 0.00 | 0.00 | 238.54 | **89.36**$^*$ |
| | 6 | 0.00 | 0.00 | 656.93 | **157.55**$^*$ |
| LSTM-s2s att. | 36 | 0.00 | 0.00 | **62.34**$^*$ | 70.92 |
| | 24 | 0.00 | 0.00 | 263.33 | **157.82**$^*$ |
| | 6 | 0.00 | 0.00 | 659.85 | **164.43**$^*$ |
| CNN-s2s | 36 | 0.75 | 0.00 | **1.44**$^*$ | 49.92 |
| | 24 | 0.13 | 0.00 | **13.75**$^*$ | 84.55 |
| | 6 | 0.00 | 0.00 | 131.63 | **29.66**$^*$ |
| Transformer | 36 | 0.00 | 0.82 | 147.83 | **6.36**$^*$ |
| | 24 | 0.00 | 0.35 | 586.22 | **26.46**$^*$ |
| | 6 | 0.00 | 0.00 | 1235.01 | **53.91**$^*$ |

(d) Composition-or-Memorization

Table 1: FPA measures the fraction of seeds that generalize according to a particular rule. Description length $L$ is averaged across examples and seeds. The lowest $L$ are in bold and $*$ denotes stat. sig. difference in $L$ ($p < 10^{-3}$, paired t-test).

toward `count`. Surprisingly, for $l \geq 30$, most learner instances show sufficiently strong inductive biases to infer *perfectly* the non-trivial `count` hypothesis. With $l = 40$, 99% of random seeds of LSTM-s2s att. and *all* (100%) of LSTM-s2s no att. seeds generalized perfectly to `count`.

Further, we see that if $L$ shows similar trends, it has a higher sensitivity. For example, while both LSTM-based learners have a similar FPA with $l = 40$, $L$ demonstrates that LSTM-s2s no att. has a stronger `count` bias.

**Add-or-Multiply** In this task, we examine learners' generalization after training on the single example $a^l \rightarrow b^{2l}$. We vary $l \in \{5, 10, 15, 20\}$. In Table 1b, we report FPA and $L$ for the three generalization hypotheses, `add`, `mul`, and `mem`. We observe, similarly to the previous task, that CNN-s2s and Transformer learners always converge perfectly to memorization.

In contrast, LSTM-based learners show non-trivial generalizations. Examining first LSTM-s2s att., when $l$=5, we note that `mem` has a high FPA and an $L$ considerably lower than others. This is consistent with the learner's behavior in the *Count-or-Memorization* task. As we increase $l$, more interesting behavior emerges. First, $L$-`mem` decreases as $l$ increases. Second, `mul`-type preference increases with $l$. Finally, $L$-`add` presents a U-shaped function of $l$. That is, for the medium example length $l$, the majority of learners switch to approximating the `add` rule (for $l = 10$). However, when $l$ grows further, a considerable fraction of these learners start to follow a `mul`-type rule. Strikingly, 98% of LSTM-s2s att. seeds generalized perfectly to the non-trivial `mul` rule. As for LSTM-s2s no att., we do not observe a strong bias to infer any of the rules when $l$=5. However, when increasing $l$, the LSTM-s2s no att. behaves similarly to LSTM-s2s att.: at first it has a preference for `add` (FPA-`add`=0.95, for $l$=10) then for `mul` (e.g. FPA-`mul`=0.94, for $l$=20).

**Hierarchical-or-Linear** We look now at learners' preference for either `hierar` or `linear` generalizations. The architectures we use were only able to consistently learn the training examples with the depth $d$ not higher than 4. Hence, in this experiment, we set $d$ to 4.

We report in Table 1c the learners' FPA and $L$. We observe that CNN-s2s exhibits a strikingly different bias compared to all other learners with a perfect agreement with the `linear` rule. In contrast, Transformer learners show a clear preference for `hierar` with a high FPA (0.69) and a low $L$ (1.21). Surprisingly, this preference increases with the embedding size and Transformers with embedding size $\geq 64$ admit an FPA-`hierar` of 1.00 (see Appendix for more details). LSTM-s2s att. learners demonstrate also a similar preference for `hierar` with an FPA of 0.30 and a considerably lower $L$ than $L$-`hierar`. Finally, while only 5% of LSTM-s2s no att. instances generalized to perfect `hierar` (and none to `linear`), $L$ confirms their preference for the `hierar` hypothesis.

**Composition-or-Memorization** In this task, we set the number of primitives $N = 40$ and vary the number of compositional examples $M \in \{6, 24, 36\}$ seen during *training*. Results are reported in Table 1d. First, we observe that FPA is only informative for CNN-s2s when trained with a large $M$. Indeed, for $M = 6$, CNN-s2s does not infer any of the candidate rules. However, according to description length $L$, we note a significant preference for `mem` over `comp`. More compositional examples CNN-based learners see at training, more biased they become toward `comp`. The remaining learners have *zero* FPA for both candidate rules. However, according to description length, LSTM-based learners have preferences similar to CNN-s2s, although weaker. That is, they show a preference for `mem` for low $M$, that declines in favor of `comp` as $M$ increases. In contrast, Transformers show a strong bias for `mem` with all tested $M$.

Overall, across all the above experiments, we see that seq2seq learners demonstrate strikingly different biases. In many cases, these biases lead to non-trivial generalizations when facing ambiguity in the training data. This spans tasks that probe for memorization, arithmetic, hierarchical, and compositional reasoning. We found that a single example is sufficient for LSTM-based learners to learn counting, addition, and multiplication. Moreover, within the same task, they can switch from one explanation to another, depending on the training example length, with *Addition-or-Multiplication* being the task where this switch happens twice. In contrast, CNN-s2s and Transformers show a strong bias toward memorization. Furthermore, all learners except for CNN-s2s demonstrate a strong bias toward the hierarchical behavior. In the task of compositional generalization, CNN-s2s shows a strong bias toward compositional reasoning that appears after a few compositional training examples. On the other hand, Transformers show a preference for memorization over compositional generalization.

We see that the conclusions derived from comparing the description length of the candidate rules are in agreement with the results under accuracy-based metrics, but provide a more nuanced picture.

**Robustness to hyper-parameters** We observe that learners' biases depend, in many cases, on the length/number of the input examples. In Appendix, we examine impact of other hyper-parameters. In particular, we study impact of (1) learners' architecture size, by varying the number of layers, hidden and embedding sizes, and (2) dropout probability. Our results show that in some cases a learner's architecture size can influence the strength of inductive biases, but rarely modify them: among the 136 tested settings, we observe only 3 cases of switching the preferences. We also found, in line with (Arpit et al., 2017), that large dropout probabilities can prevent `mem`-type generalization.
Finally, in Appendix we show that a variant of Transformer learners, namely the joint source-target self-attention learner (He et al., 2018; Fonollosa et al., 2019), displays the same preferences as the standard Transformer learners. This variant resembles the "decoder-only" architecture used in language modeling (Radford et al., 2019; Brown et al., 2020). This result demonstrates that our tasks and bias measures could be applied for studying inductive biases of language model architectures.

## 6 RELATED WORK

Dessì & Baroni (2019) found that, unlike LSTM-s2s learners, CNN-s2s can perform compositional generalization on SCAN. Our experiments indicate that this only happens when enough compositional examples are provided in the training. Moreover, in such a case, attention-enabled LSTM-s2s also start to prefer compositional generalization over memorization.

McCoy et al. (2020) studied inductive biases of recurrent neural architectures in two synthetic tasks, English question formation and tense inflection. They found that only tree-based architectures show robust preference for hierarchical reasoning, in contrast to LSTM-s2s learners that generalized linearly. Our experiments on the hyperparameter robustness, reported in Appendix, indicate that the preferences over linear/hierarchical reasoning are strongly affected by the dropout probability, with learners shifting to linear behavior with low probabilities. As McCoy et al. (2020) experimented with a low dropout probability of 0.1, we believe this explains the misalignment of the conclusions.

Overall, our study shows that inductive biases are more complicated than it seemed in these prior works and a more careful analysis is crucial. We believe that our extremely controlled setup with very little confounds is a good addition to those studies.

Another line of research investigates theoretically learners' capabilities, that is, the classes of the hypothesis that a learner can discover (Siegelmann & Sontag, 1992; Suzgun et al., 2019; Merrill et al., 2020). For example, Weiss et al. (2018) demonstrated that LSTM cells can count. In turn, we demonstrate that LSTM-s2s learners are not only capable but also *biased* toward arithmetic behavior.

## 7 DISCUSSION AND CONCLUSION

In this work, we studied inductive biases of standard seq2seq learners, Transformer-, LSTM-, and CNN-based. To do so, we introduced four new tasks, which allowed us to cover an interesting spectrum of behaviors useful for language learning. In particular, we considered arithmetic, hierarchical, and compositional "reasoning". Next, we connected the problem of finding and measuring inductive biases to Solomonoff's theory of induction and proposed to use a dataset's description length under a learner as a tool for sensitive measurement of inductive biases.

In our experiments, we found that the seq2seq learners have strikingly different inductive biases and some of them generalize non-trivially when facing ambiguity. For instance, a single training example is sufficient for LSTM-based learners to learn perfectly how to count, to add and to multiply by a constant. Transformers and, to a lesser degree, LSTM-s2s demonstrated preferences for the hierarchical bias, a bias that has been argued to govern children's acquisition of syntax. Interestingly, such biases arose with no explicit wiring for them. Our results support then Elman et al. (1998)'s theory which states that human's inductive biases can arise from low-level architectural constraints in the brain with no need for an explicit encoding of a linguistic structure. However, how the brain, or, more generally, a learner is wired to admit a specific inductive bias is still an important open question.

Across our experiments, we also observed that description length is consistent with "intuitive" measurements of inductive biases, and, at the same time, it turned out to be more sensitive. This also indicates that, in the presence of ambiguity in the training data, a learner is more likely to follow the alternative with the shorter description length (i.e. the simplest one) when applied on unseen data, showing consistency with the prescriptions of the theory of induction (Solomonoff, 1964). A similar simplicity preference is argued to play a role in human language acquisition (Perfors et al., 2011).

Our work provides simple tools to investigate learners' biases. We first show that FPA is an intuitive measure to study biases when provided with simple tasks. Second, we present description length as a robust measure to fairly compare learners' biases. This metric considers learners' size and their ease of learning as opposed to accuracy-based metrics. Besides, it is a model- and task-agnostic measure that succeeds in unveiling learners' biases even when presented with more complex tasks with spurious correlations.

Our findings can guide for architecture selection in the low-data regimes where inductive biases might have a higher influence on model's generalization performance. Large sparse datasets can also benefit from predictable behavior in few-shot scenarios akin to what we consider.

Finally, our results demonstrate that relatively large deep learning models *can* generalize non-trivially from as little as one example – as long as the task is aligned with the their inductive biases. We believe this should reinforce interest in future work on injecting useful inductive biases in our learners and, we hope, our findings and setup can provide a fertile ground for such work.

## ACKNOWLEDGEMENTS

The authors are grateful to Marco Baroni, Emmanuel Dupoux, Emmanuel Chemla and participants of the EViL seminar for their feedback on our work.

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

| | length $l$ | FPA | | | | $L$, nats | | | |
|---|---|---|---|---|---|---|---|---|---|
| | | mul1 | mul2 | mul3 | mem | mul1 | mul2 | mul3 | mem |
| LSTM-s2s no att. | 20 | 0.00 | 0.01 | 0.78 | 0.00 | 52.27 | 22.20 | **1.17**$^*$ | 77.75 |
| | 15 | 0.00 | 0.13 | 0.45 | 0.00 | 40.46 | 13.22 | **6.14**$^*$ | 66.10 |
| | 10 | 0.00 | 0.92 | 0.00 | 0.00 | 26.48 | **0.65**$^*$ | 22.26 | 53.81 |
| | 5 | 0.49 | 0.00 | 0.00 | 0.00 | **1.97**$^*$ | 26.50 | 54.97 | 36.13 |
| LSTM-s2s att. | $20^5$ | 0.00 | 0.19 | 0.62 | 0.00 | 36.76 | 20.35 | **9.35**$^*$ | 49.34 |
| | $15^6$ | 0.00 | 0.14 | 0.76 | 0.00 | 37.84 | 18.42 | **5.53**$^*$ | 56.43 |
| | 10 | 0.02 | 0.45 | 0.49 | 0.00 | 29.83 | 11.67 | **8.96** | 45.47 |
| | 5 | 0.01 | 0.03 | 0.00 | 0.64 | 32.97 | 48.26 | 60.38 | **2.44**$^*$ |
| CNN-s2s | 20 | 0.00 | 0.00 | 0.00 | 0.99 | 263.82 | 262.01 | 261.71 | **0.02**$^*$ |
| | 15 | 0.00 | 0.00 | 0.00 | 1.00 | 250.97 | 253.32 | 253.08 | **0.00**$^*$ |
| | 10 | 0.00 | 0.00 | 0.00 | 1.00 | 243.17 | 245.16 | 248.28 | **0.00**$^*$ |
| | 5 | 0.00 | 0.00 | 0.00 | 1.00 | 258.10 | 257.79 | 264.06 | **0.00**$^*$ |
| Transformer | 20 | 0.00 | 0.00 | 0.00 | 0.97 | 37.90 | 51.47 | 57.57 | **5.31**$^*$ |
| | 15 | 0.00 | 0.00 | 0.00 | 1.00 | 40.36 | 51.62 | 57.42 | **2.50**$^*$ |
| | 10 | 0.00 | 0.00 | 0.00 | 1.00 | 38.05 | 49.88 | 55.61 | **2.47**$^*$ |
| | 5 | 0.00 | 0.00 | 0.00 | 1.00 | 37.96 | 51.83 | 60.19 | **0.74**$^*$ |

Table 2: Multiplication by 3. FPA measures the fraction of seeds that generalize according to a particular rule. Description length $L$ is averaged across examples and seeds. The lowest $L$ are in bold and $*$ denotes stat. sig. difference in $L$ ($p < 10^{-3}$, paired t-test).

# A    CAN SEQ2SEQ LEARNERS MULTIPLY BY 3?

In our experiments on the *Multiply-or-Add* task we saw that LSTM-s2s learners are able to learn to multiply by 2 from a single example. A natural further question is whether these learners can learn to multiply by larger numbers? Here we only provide a preliminary study in a hope to inspire more focused studies.

We build a task that is similar to *Multiply-or-Add*, but centered around multiplication by 3 instead of 2. The single training example represents a mapping of an input string $a^l$ to an output string $b^{3l}$. As test inputs, we use $a^m$ with $m$ coming from an interval $[l-3, l+3]$.

Since $3x$ can be represented as a combination of addition and multiplication in several ways ($3x = x + 2x = 2x + x$), we consider 4 different candidate rules. As before, by mem we denote the constant mapping from $a^m \rightarrow b^{3l}$. mul1 represents the mapping $a^m \rightarrow b^{2l+m}$. mul2 corresponds to $a^m \rightarrow b^{l+2m}$ and mul3 denotes $a^m \rightarrow b^{3m}$. The "explanation" mul1 is akin to the add rule in the *Multiply-or-Add* task. We use the same hyperparameters and training procedure described in Section 4.

We report the results in Table 2. Like the results observed in the *Multiply-or-Add* task, both CNN-s2s and Transformer learners show a strong preference for the mem rule while LSTM-based learners switch their generalization according to the length of the training example $l$. Indeed, for CNN-s2s and Transformer, we note an FPA-mem>0.97 independently of $l$ (with $L$-mem significantly lower than others). LSTM-s2s att. learners start inferring the mem rule for $l = 5$ (FPA=0.64, $L$=2.44), then switch to comparable preference for mul2 and mul3 when $l = 10$, and finally show a significant bias toward the mul3 hypothesis for $l \in \{15, 20\}$ (e.g. FPA-mul3=0.76 for $l = 15$). LSTM-s2s no att. learners are also subject to a similar switch of preference. That is, for $l = 5$, these learners have a significant bias toward mul1 (FPA=0.49). Strikingly, when $l = 10$, 92% LSTM-s2s no att. learners inferred perfectly the mul2 rule after one training example. Finally, we observe again another switch to approximate mul3 for $l \in \{15, 20\}$.

Overall, while CNN-s2s and Transformer learners show a significant and robust bias toward mem, LSTM-based learners generalize differently depending on the training input length. In particular, our results suggest that these learners avoid adding very large integers by switching to the multiplicative explanation in those cases. Answering our initial question, we see that LSTM-based learners *can* learn to multiply by 3 from a single example.

---

[5]Only 21% of learners succeeded in learning the training example in this setting.
[6]Only 51% of learners succeeded in learning the training example in this setting.

# B  ROBUSTNESS TO CHANGES IN ARCHITECTURE

In this section, we examine how changing different hyper-parameters affects learners' preferences for memorization, arithmetic, hierarchical and compositional reasoning. In particular, we vary the number of layers, hidden and embedding sizes of the different learners and test their generalization on *Count-or-Memorization*, *Add-or-Multiply*, *Hierarchical-or-Linear*, and *Composition-or-Memorization* tasks.

In all the experiments, we fix $l = 40$ and $l = 20$ for *Count-or-Memorization* and *Add-or-Multiply* respectively, $d = 4$ for *Hierarchical-or-Linear* and $M = 36$ for *Composition-or-Memorization*. Finally, we keep the same training and evaluation procedure as detailed in Section 4.2 of the main text. However, we use 20 different random seeds instead of 100.

## B.1  NUMBER OF HIDDEN LAYERS ($N_{Layer}$)

We experiment with the standard seq2seq learners described in the main paper and vary the number of layers $N_{Layer} \in \{1, 3, 10\}$. Results are reported in Table 3.

First, when looking at the interplay between mem and count (Table 3a), we observe that, independently of $N_{Layer}$, more than 97% of CNN-s2s and Transformer learners inferred *perfectly* the mem rule (i.e. output $b^{40}$ for any given input $a^m$). Further, if the preference for count decreases with the increase of $N_{Layer}$, LSTM-s2s att. learners display in all cases a significant bias toward count with a large FPA and an $L$ significantly lower than $L$-mem . However, we note a decline of the bias toward count when considering LSTM-s2s no att. learners. For $N_{Layer} = 1$, 100% of the seeds generalize to perfect count, versus 6% for $N_{Layer} = 10$. Note that this lower preference for count is followed by an increase of preference for mem. However, there is no significant switch of preferences according to $L$.

Second, we consider the *Add-or-Multiply* task where we examine the three generalization hypothesis add, mul and mem. Results are reported in Table 3b. Similarly to the previous task, Transformer and CNN-s2s learners are robust to the number of layers change. They perfectly follow the mem rule with FPA-mem=1.00. However, LSTM-based learners show a more complex behavior: If single-layer LSTM-s2s att. and no att. demonstrate a considerable bias for mul (FPA-mul>0.94), this bias fades in favor of add and memo for larger architectures.

Third, in Table 3c, we observe that larger learners are slightly less biased toward hierar and linear. However, we do not observe any switch of preferences. That is, across different $N_{layers}$, CNN-s2s learners prefer the linear rule, whereas Transformers and, to a lesser degree, LSTM-based learners show a significant preference for the hierar rule.

Finally, we do not observe an impact of $N_{Layer}$ on the *Composition-or-Memorization* task (see Table 3d), demonstrating the robustness of biases w.r.t. learners' size.

In sum, we note little impact of $N_{Layer}$ on learners' generalizations. Indeed, *LSTM-based learners can learn to perform counting from a single training example, even when experimenting with 10-layer architectures.* For most tested $N_{Layer}$, they favor mul and add over mem. In contrast, Transformer and CNN-s2s perform systematic memorization of the single training example. Furthermore, independently of $N_{Layer}$, Transformer and LSTM-based learners show a bias toward the hierar hypothesis over the linear one, while CNN-based learners do the opposite. Finally, the compositional experiments further support how inductive biases are barely influenced by the change of the number of layers.

## B.2  HIDDEN SIZE ($S_{Hidden}$)

We experimented in the main text with standard seq2seq learners when hidden size $S_{Hidden} = 512$. In this section, we look at the effect of $S_{Hidden}$ varying it in $\{128, 512, 1024\}$. We report learners performances in Table 4.

First, Table 4a demonstrates how minor effect hidden size has on learners counting/memorization performances. Indeed, for any given $S_{Hidden}$, between 84% and 100% of LSTM-based learners learn perfect counting after only one training example. Similarly, and with even a lower variation, more than 91% of Transformer and CNN-s2s learners memorize the single training example outputting $b^{40}$ for any given $a^m$.

The same observation can be made when studying the interplay between the hierar and linear biases (Table 4c) and comp and mem biases (Table 4d). Concretely, Tables 4c and 4d show that learners' generalizations are stable across $S_{Hidden}$ values with the exception of LSTM-based learners that loose significance for some tested $S_{Hidden}$ values. Yet, there is no switch of preference.

Finally, as demonstrated in Table 4b, *all* Transformer and CNN-s2s learners perform perfect memorization when tested on the *Add-or-Multiply* task independently of their hidden sizes. Both LSTM-based learners are significantly biased toward mul for $S_{Hidden} \in \{512, 1024\}$. However, when experimenting with smaller

| | $N_{Layer}$ | FPA count | mem | $L$, nats count | mem |
|---|---|---|---|---|---|
| LSTM-s2s no att. | 1 | 1.00 | 0.00 | **0.01**$^*$ | 97.51 |
| | 3 | 0.80 | 0.00 | **0.00**$^*$ | 60.80 |
| | 10 | 0.06 | 0.47 | **16.89** | 22.02 |
| LSTM-s2s att. | 1 | 0.99 | 0.00 | **7.84**$^*$ | 121.48 |
| | 3 | 0.50 | 0.00 | **11.30**$^*$ | 57.57 |
| | 10 | 0.39 | 0.22 | **22.86**$^*$ | 45.13 |
| CNN-s2s | 1 | 0.00 | 0.98 | 660.73 | **0.02**$^*$ |
| | 3 | 0.00 | 1.00 | 1685.18 | **0.00**$^*$ |
| | 10 | - | - | - | - |
| Transformer | 1 | 0.00 | 0.97 | 116.34 | **11.10**$^*$ |
| | 3 | 0.00 | 1.00 | 139.58 | **0.31**$^*$ |
| | 10 | - | - | - | - |

(a) Count-or-Memorization

| | $N_{Layer}$ | FPA add | mul | mem | $L$, nats add | mul | mem |
|---|---|---|---|---|---|---|---|
| LSTM-s2s no att. | 1 | 0.00 | 0.94 | 0.00 | 25.42 | **0.31**$^*$ | 57.32 |
| | 3 | 0.15 | 0.20 | 0.00 | **9.82** | 10.11 | 33.16 |
| | 10 | 0.28 | 0.00 | 0.33 | **5.83** | 23.16 | 10.01 |
| LSTM-s2s att. | 1 | 0.00 | 0.98 | 0.00 | 30.26 | **1.40**$^*$ | 58.84 |
| | 3 | 0.65 | 0.20 | 0.00 | **5.88**$^*$ | 15.67 | 27.82 |
| | 10 | 0.11 | 0.11 | 0.28 | 9.66 | 27.72 | **8.26** |
| CNN-s2s | 1 | 0.00 | 0.00 | 1.00 | 318.12 | 346.19 | **0.00**$^*$ |
| | 3 | 0.00 | 0.00 | 1.00 | 1058.27 | 824.93 | **2.31**$^*$ |
| | 10 | - | - | - | - | - | - |
| Transformer | 1 | 0.00 | 0.00 | 1.00 | 38.77 | 50.64 | **3.50**$^*$ |
| | 3 | 0.00 | 0.00 | 1.00 | 40.03 | 57.73 | **0.18**$^*$ |
| | 10 | - | - | - | - | - | - |

(b) Add-or-Multiply

| | $N_{Layer}$ | FPA hierar | linear | $L$, nats hierar | linear |
|---|---|---|---|---|---|
| LSTM-s2s no att. | 1 | 0.05 | 0.00 | **31.04**$^*$ | 61.84 |
| | 3 | 0.00 | 0.00 | **29.08**$^*$ | 50.92 |
| | 10 | - | - | - | - |
| LSTM-s2s att. | 1 | 0.30 | 0.00 | **26.32**$^*$ | 57.20 |
| | 3 | 0.00 | 0.00 | **32.32**$^*$ | 50.12 |
| | 10 | - | - | - | - |
| CNN-s2s | 1 | 0.00 | 1.00 | 202.64 | **0.00**$^*$ |
| | 3 | 0.00 | 0.70 | 222.32 | **1.84**$^*$ |
| | 10 | - | - | - | - |
| Transformer | 1 | 0.69 | 0.00 | **4.84**$^*$ | 35.04 |
| | 3 | 0.56 | 0.00 | **5.48**$^*$ | 29.16 |
| | 10 | - | - | - | - |

(c) Hierarchical-or-Linear

| | $N_{Layer}$ | FPA comp | mem | $L$, nats comp | mem |
|---|---|---|---|---|---|
| LSTM-s2s no att. | 1 | 0.00 | 0.00 | 42.65 | **38.55** |
| | 3 | 0.00 | 0.00 | 37.01 | **34.16** |
| | 10 | - | - | - | - |
| LSTM-s2s att. | 1 | 0.00 | 0.00 | **62.34**$^*$ | 70.92 |
| | 3 | 0.00 | 0.00 | **46.10**$^*$ | 58.79 |
| | 10 | - | - | - | - |
| CNN-s2s | 1 | 0.75 | 0.00 | **1.44**$^*$ | 49.92 |
| | 3 | 0.95 | 0.00 | **1.79**$^*$ | 66.44 |
| | 10 | - | - | - | - |
| Transformer | 1 | 0.00 | 0.00 | 147.83 | **6.36**$^*$ |
| | 3 | - | - | - | - |
| | 10 | - | - | - | - |

(d) Composition-or-Memorization

Table 3: **Effect of the number of layers** ($N_{Layer}$): FPA measures the fraction of seeds that generalize according to a particular rule. Description length $L$ is averaged across examples and seeds. The lowest $L$ are in bold and $*$ denotes stat. sig. difference in $L$ ($p < 10^{-2}$, paired t-test.). '-' denotes settings where learners have lower than 70% success rate on the train set.

$S_{Hidden}$ (=128), we detect a switch of preference for LSTM-s2s no att. learners. The latter start approximating `add`-type rule (with significantly lower $L$). Lastly, we do not distinguish any significant difference between `add` and `mul` for LSTM-s2s att. when $S_{Hidden} = 128$.

Taken together, learners' biases are quite robust to $S_{Hidden}$ variations. We however note a switch of preference from `mul` to `add` for LSTM-s2s no att. learners when decreasing $S_{Hidden}$. Furthermore, we see a loss of significant preference in five distinct settings.

## B.3   EMBEDDING SIZE ($S_{Emb}$)

We look here at the effect of the embedding size, $S_{Emb}$, on learners' generalizations. In particular, we vary $S_{Emb} \in \{16, 64, 256\}$. Results are reported in Table 5.

Across sub-tables 5a, 5b and 5c, we see small influence of $S_{Emb}$ on learners' biases. For example, if we consider the *Count-or-Memorization* task when varying $S_{Emb}$ (see Table 5a), between 95% and 100% of LSTM-s2s no att. learners inferred perfectly the `count` hypothesis. More striking, between 99% and 100% of LSTM-s2s att. learners learned the `count` rule after one training example. The same trend is observed for the remaining learners and across the other tasks; *Add-or-Multiply* (Table 5b) and *Hierarchical-or-Linear* (Table 5c). Yet, we still discern in some cases, systematic, but low, effects of $S_{Emb}$. First, the larger $S_{Emb}$ is, the lower FPA-`mul` of LSTM-s2s no att. learners is (from 0.94 for $S_{Emb} = 16$ to 0.84 for $S_{Emb} = 256$). However, LSTM-s2s no att. learners still have considerable preference for `mul` for any tested $S_{Emb}$. Second, we see an increase of Transformer's preference for `hierar` with the increase of $S_{Emb}$. Surprisingly, for $S_{Emb} \geq 64$, 100% of Transformer learners generalize to perfect `hierar` hypothesis.

Finally, when considering the *Composition-or-Memorization* task, we observe in Table 5d, for all learners, a tendency to prefer `comp` generalization when $S_{Emb}$ is large. For example, if, for LSTM-based learners, we note 0.00 `comp`-FPA when $S_{Emb} = 16$, `comp`-FPA$\geq 0.85$ with $S_{Emb} = 256$. Interestingly, Transformers switch their preference, showing a significant bias toward compositional reasoning with $S_{Emb} = 256$.

In this section, we studied the impact of the number of layers, hidden and embedding sizes on learners' generalizations. We found that, if these hyper-parameters can influence, in some cases, the degree of one learner's preference w.r.t. a given rule, inductive biases are quite robust to their changes. In particular, among all tested combinations, we observe only 3 cases of preference switch (out of 136).

## C   ROBUSTNESS TO CHANGES IN TRAINING PARAMETERS

We examine here the effect of the training parameters on learners' biases. As previously, we only consider the *Count-or-Memorization* task with $l = 40$, the *Add-or-Multiply* task with $l = 20$, the *Hierarchical-or-Linear* task with $d = 4$ and the *Composition-or-Memorization* task with $M = 36$. We experiment with the architectures detailed in the main text; however, we use 20 different random seeds instead of 100.

We consider in this section two different hyperparameters: (1) the choice of the optimizer, and (2) the dropout probability.

**Optimizer** We experiment with replacing the Adam optimizer (Kingma & Ba, 2014) with SGD (Bottou et al., 2018). We found experimentally that learners failed to learn the training examples consistently in most of the settings. Yet, when successful, they showed the same preferences. In particular, Transformer and CNN-s2s were the only learners that had good performances on *Count-or-Memorization* and *Add-or-Multiply* train sets (success rate higher than 70%). These learners showed a prefect generalization to the `mem` rule in both tasks. Prior works had shown that SGD leads to better generalization compared to other adaptive variants (Wilson et al., 2017). However, based on our few successful instances with SGD, we do not observe a difference between SGD and Adam results. One main difference compared to the prior work is the definition of generalization. For instance, in the *Count-or-Memorization* task, memorizing the training example could be seen as a perfect generalization as it explains the training set fully. Hence, preferring memorization when trained with SGD, as opposed to counting, does not contradict the advantage of SGD when testing generalization performances.

**Dropout** We then examine how the dropout probability affects learners' preferences. We use, as mentioned in the main paper, Adam optimizer and vary the dropout probability $dropout \in \{0.0, 0.2, 0.5\}$. Results are reported in Table 6.

Both *Count-or-Memorization* (Table 6a) and *Add-or-Multiply* (Table 6b) tasks show the same trend. First, Transformer and CNN-s2s learners prefer consistently the `mem` rule. Second, when looking at LSTM-based learners, we distinguish a more complex behavior. For $dropout \geq 0.2$, LSTM-based learners show a significant preference for arithmetic reasoning (`count` for the *Count-or-Memorization* task and `mul` for the *Add-or-Multiply* task). However, when $dropout = 0.0$, we see different preferences. In particular, both LSTM-based learners show a preference for `mem` (not significant for LSTM-s2s att.), LSTM-s2s no att. learners are significantly

| | | FPA | | $L$, nats | |
|---|---|---|---|---|---|
| | $S_{Hidden}$ | count | mem | count | mem |
| LSTM-s2s no att. | 128 | 0.84 | 0.00 | **8.25**$^*$ | 109.84 |
| | 512 | 1.00 | 0.00 | **0.01**$^*$ | 97.51 |
| | 1024 | 1.00 | 0.00 | **0.00**$^*$ | 149.86 |
| LSTM-s2s att. | 128 | 0.90 | 0.00 | **0.01**$^*$ | 89.31 |
| | 512 | 0.99 | 0.00 | **7.84**$^*$ | 121.48 |
| | 1024 | 1.00 | 0.00 | **0.00**$^*$ | 300.93 |
| CNN-s2s | 128 | 0.00 | 1.00 | 805.01 | **0.00**$^*$ |
| | 512 | 0.00 | 0.98 | 660.73 | **0.02**$^*$ |
| | 1024 | 0.00 | 1.00 | 993.85 | **0.00**$^*$ |
| Transformer | 128 | 0.00 | 0.91 | 110.68 | **9.57**$^*$ |
| | 512 | 0.00 | 0.97 | 116.34 | **11.10**$^*$ |
| | 1024 | 0.00 | 0.94 | 122.38 | **1.42**$^*$ |

(a) Count-or-Memorization

| | | FPA | | | $L$, nats | | |
|---|---|---|---|---|---|---|---|
| | $S_{Hidden}$ | add | mul | mem | add | mul | mem |
| LSTM-s2s no att. | 128 | 0.00 | 0.00 | 0.00 | **7.56**$^*$ | 19.66 | 43.72 |
| | 512 | 0.00 | 0.94 | 0.00 | 25.42 | **0.31**$^*$ | 57.32 |
| | 1024 | 0.25 | 0.75 | 0.00 | 30.29 | **5.26**$^*$ | 77.99 |
| LSTM-s2s att. | 128 | 0.00 | 0.00 | 0.00 | **15.32** | 17.37 | 45.09 |
| | 512 | 0.00 | 0.98 | 0.00 | 30.26 | **1.40**$^*$ | 58.84 |
| | 1024 | 0.00 | 1.00 | 0.00 | 51.70 | **3.09**$^*$ | 86.82 |
| CNN-s2s | 128 | 0.00 | 0.00 | 1.00 | 281.34 | 301.75 | **0.02**$^*$ |
| | 512 | 0.00 | 0.00 | 1.00 | 318.12 | 346.19 | **0.00**$^*$ |
| | 1024 | 0.00 | 0.00 | 1.00 | 520.75 | 508.75 | **0.00**$^*$ |
| Transformer | 128 | 0.00 | 0.00 | 1.00 | 35.21 | 46.07 | **2.94**$^*$ |
| | 512 | 0.00 | 0.00 | 1.00 | 38.77 | 50.64 | **3.50**$^*$ |
| | 1024 | 0.00 | 0.00 | 1.00 | 38.74 | 51.71 | **0.88**$^*$ |

(b) Add-or-Multiply

| | | FPA | | $L$, nats | |
|---|---|---|---|---|---|
| | $S_{Hidden}$ | hierar | linear | hierar | linear |
| LSTM-s2s no att. | 128 | 0.00 | 0.00 | **30.40**$^*$ | 79.00 |
| | 512 | 0.05 | 0.00 | **31.04**$^*$ | 61.84 |
| | 1024 | 0.00 | 0.00 | 60.24 | **47.24** |
| LSTM-s2s att. | 128 | 0.00 | 0.00 | **32.72**$^*$ | 72.28 |
| | 512 | 0.30 | 0.00 | **26.32**$^*$ | 57.2 |
| | 1024 | 0.05 | 0.00 | **65.80** | 73.80 |
| CNN-s2s | 128 | 0.00 | 0.95 | 178.88 | **0.00**$^*$ |
| | 512 | 0.00 | 1.00 | 202.64 | **0.00**$^*$ |
| | 1024 | 0.00 | 0.95 | 225.36 | **0.12**$^*$ |
| Transformer | 128 | 0.75 | 0.00 | **2.96**$^*$ | 36.92 |
| | 512 | 0.69 | 0.00 | **4.84**$^*$ | 35.04 |
| | 1024 | 0.75 | 0.00 | **31.04**$^*$ | 61.84 |

(c) Hierarchical-or-Linear

| | | FPA | | $L$, nats | |
|---|---|---|---|---|---|
| | $S_{Hidden}$ | comp | mem | comp | mem |
| LSTM-s2s no att. | 128 | 0.00 | 0.00 | **37.67**$^*$ | 58.52 |
| | 512 | 0.00 | 0.00 | 42.65 | **38.55** |
| | 1024 | 0.00 | 0.00 | **37.66**$^*$ | 79.71 |
| LSTM-s2s att. | 128 | 0.00 | 0.00 | 66.65 | **63.84** |
| | 512 | 0.00 | 0.00 | **62.34**$^*$ | 70.92 |
| | 1024 | 0.00 | 0.00 | **42.38**$^*$ | 73.28 |
| CNN-s2s | 128 | 0.85 | 0.00 | **0.99**$^*$ | 53.38 |
| | 512 | 0.75 | 0.00 | **1.44**$^*$ | 49.92 |
| | 1024 | 0.80 | 0.00 | **1.24**$^*$ | 54.86 |
| Transformer | 128 | 0.00 | 0.66 | 155.50 | **6.23**$^*$ |
| | 512 | 0.00 | 0.00 | 147.83 | **6.36**$^*$ |
| | 1024 | 0.00 | 0.60 | 151.70 | **6.02**$^*$ |

(d) Composition-or-Memorization

Table 4: **Effect of the hidden size** ($S_{Hidden}$): FPA measures the fraction of seeds that generalize according to a particular rule. Description length $L$ is averaged across examples and seeds. The lowest $L$ are in bold and $*$ denotes stat. sig. difference in $L$ ($p < 10^{-2}$, paired t-test).

| | $S_{Emb}$ | FPA count | FPA mem | $L$, nats count | $L$, nats mem |
|---|---|---|---|---|---|
| LSTM-s2s no att. | 16 | 1.00 | 0.00 | **0.01**\* | 97.51 |
| | 64 | 0.95 | 0.00 | **0.00**\* | 91.54 |
| | 256 | 1.00 | 0.00 | **0.00**\* | 90.32 |
| LSTM-s2s att. | 16 | 0.99 | 0.00 | **7.84**\* | 121.48 |
| | 64 | 1.00 | 0.00 | **11.12**\* | 117.39 |
| | 256 | 1.00 | 0.00 | **9.79**\* | 127.31 |
| CNN-s2s | 16 | 0.00 | 0.98 | 660.73 | **0.02**\* |
| | 64 | 0.00 | 1.00 | 670.53 | **0.00**\* |
| | 256 | 0.00 | 0.95 | 826.23 | **0.01**\* |
| Transformer | 16 | 0.00 | 0.97 | 116.34 | **11.10**\* |
| | 64 | 0.00 | 1.00 | 232.53 | **0.00**\* |
| | 256 | 0.00 | 1.00 | 338.88 | **0.00**\* |

(a) Count-or-Memorization

| | $S_{Emb}$ | FPA add | FPA mul | FPA mem | $L$, nats add | $L$, nats mul | $L$, nats mem |
|---|---|---|---|---|---|---|---|
| LSTM-s2s no att. | 16 | 0.00 | 0.94 | 0.00 | 25.42 | **0.31**\* | 57.32 |
| | 64 | 0.05 | 0.90 | 0.00 | 23.70 | **1.41**\* | 51.33 |
| | 256 | 0.05 | 0.84 | 0.00 | 20.42 | **1.33**\* | 51.34 |
| LSTM-s2s att. | 16 | 0.00 | 0.98 | 0.00 | 30.26 | **1.40**\* | 58.84 |
| | 64 | 0.07 | 0.93 | 0.00 | 26.71 | **2.54**\* | 50.65 |
| | 256 | 0.00 | 1.00 | 0.00 | 26.83 | **1.94**\* | 51.20 |
| CNN-s2s | 16 | 0.00 | 0.00 | 1.00 | 318.12 | 346.19 | **0.00**\* |
| | 64 | 0.00 | 0.00 | 1.00 | 293.86 | 294.50 | **0.00**\* |
| | 256 | 0.00 | 0.00 | 1.00 | 486.81 | 447.20 | **0.00**\* |
| Transformer | 16 | 0.00 | 0.00 | 1.00 | 38.77 | 50.64 | **3.50**\* |
| | 64 | 0.00 | 0.00 | 1.00 | 87.11 | 142.83 | **0.00**\* |
| | 256 | 0.00 | 0.00 | 1.00 | 118.34 | 172.65 | **0.00**\* |

(b) Add-or-Multiply

| | $S_{Emb}$ | FPA hierar | FPA linear | $L$, nats hierar | $L$, nats linear |
|---|---|---|---|---|---|
| LSTM-s2s no att. | 16 | 0.05 | 0.00 | **31.04**\* | 61.84 |
| | 64 | 0.10 | 0.00 | **34.44**\* | 72.2 |
| | 256 | 0.00 | 0.00 | **36.32** | 76.56 |
| LSTM-s2s att. | 16 | 0.30 | 0.00 | **26.32**\* | 57.2 |
| | 64 | 0.30 | 0.00 | **23.4** | 72.68 |
| | 256 | 0.05 | 0.00 | **55.56**\* | 90.76 |
| CNN-s2s | 16 | 0.00 | 1.00 | 202.64 | **0.00**\* |
| | 64 | 0.00 | 1.00 | 227.12 | **0.08**\* |
| | 256 | 0.00 | 0.94 | 419.28 | **8.84**\* |
| Transformer | 16 | 0.69 | 0.00 | **4.84**\* | 35.04 |
| | 64 | 1.00 | 0.00 | **0.00**\* | 81.16 |
| | 256 | 1.00 | 0.00 | **0.56**\* | 121.68 |

(c) Hierarchical-or-Linear

| | $S_{Emb}$ | FPA comp | FPA mem | $L$, nats comp | $L$, nats mem |
|---|---|---|---|---|---|
| LSTM-s2s no att. | 16 | 0.00 | 0.00 | 42.65 | **38.55** |
| | 64 | 0.95 | 0.00 | **0.11**\* | 31.44 |
| | 256 | 0.95 | 0.00 | **2.12**\* | 23.63 |
| LSTM-s2s att. | 16 | 0.00 | 0.00 | **62.34**\* | 70.92 |
| | 64 | 0.95 | 0.00 | **0.36**\* | 47.57 |
| | 254 | 0.85 | 0.00 | **0.86**\* | 49.39 |
| CNN-s2s | 16 | 0.75 | 0.00 | **1.44**\* | 49.92 |
| | 64 | 1.00 | 0.00 | **0.01**\* | 112.09 |
| | 256 | 0.95 | 0.00 | **0.18**\* | 139.51 |
| Transformer | 16 | 0.00 | 0.00 | 147.83 | **6.36**\* |
| | 64 | 0.00 | 0.00 | 103.25 | **69.07**\* |
| | 256 | 0.10 | 0.00 | **70.76**\* | 107.54 |

(d) Composition-or-Memorization

Table 5: **Effect of the embedding size** ($S_{Emb}$): FPA measures the fraction of seeds that generalize according to a particular rule. Description length $L$ is averaged across examples and seeds. The lowest $L$ are in bold and \* denotes stat. sig. difference in $L$ ($p < 10^{-2}$, paired t-test).

| jump | $\Rightarrow$ | JUMP |
|---|---|---|
| jump around right | $\Rightarrow$ | RTURN JUMP RTURN JUMP RTURN JUMP RTURN JUMP |
| turn left twice | $\Rightarrow$ | LTURN LTURN |
| jump opposite left after walk around left | $\Rightarrow$ | LTURN WALK LTURN WALK LTURN WALK LTURN WALK LTURN LTURN JUMP |

Figure 2: Examples of SCAN trajectories and instructions, adopted from (Lake & Baroni, 2017).

biased toward add whereas LSTM-s2s att. do not show any significant bias with a slight preference for mem. In sum, the lower *dropout* is, the more likely learners will overfit the mem rule.

The same observation holds for the *Composition-or-Memorization* task in Table 6d. That is, *dropout* has no impact on CNN-s2s and Transformer biases (the former show a significant preference for comp while the latter prefers mem for any tested *dropout* value), but does impact LSTM-based learners' biases, such as a *dropout* = 0.0 favors memorization.

Finally, we consider the *Hierarchical-or-Linear* task (see Table 6c). We observe that, for any *dropout* value, CNN-s2s and Transformer inductive biases remain the same. Indeed, CNN-s2s learners show a consistent preference for linear with FPA$\geq$ 0.75 while Transformers prefer hierar (note that this preference is not very large for *dropout* = 0.0 with an FPA-hierar of 0.05, compared to an FPA-hierar of 1.00 and 0.69 for *dropout* = 0.2 and 0.5 respectively). On the other hand, *dropout* has a larger impact on LSTM-based learners. When *dropout* = 0.5, both LSTM-s2s prefer the hierar hypothesis. However, for *dropout* = 0.0, both learners do not show any significant preference for any of the rules (with 0 FPA for both rules and close $L$ values).

## D    SCAN

In the main text, we studied preferences of the learners towards compositional generalization in the *Composition-or-Memorization* task. Here, we resort to a larger-scale SCAN task (Lake & Baroni, 2017), which can be thought of as a generalization of *Composition-or-Memorization* with multiple modifiers. In SCAN, inputs are sequences that represent trajectories and outputs are step-wise instructions for following these instructions (see Figure 2). We experiment with the SCAN-jump split of the dataset, where the test set (7706 examples) is obtained by filtering all compositional uses of one of the primitives, jump. The train set (14670 examples) contains all uses of all other primitives (including compositional), and lines where jump occurs in isolation (only as a primitive). This makes the training data ambiguous only for the jump instruction. However, learners can transfer compositional rules across the primitives. We refer to this split simply as SCAN.

We consider two generalizations: (1) all input sequences with jump are mapped to a single instruction JUMP as memorized from the training examples, or (2) underlying compositional grammar. We call them mem and comp, respectively. We used the original test as a representative of the comp candidate explanation and generated one for mem ourselves.

Recently, Hupkes et al. (2020) compared how different seq2seq learners perform on a context-free grammar, similar to SCAN, that requires compositional reasoning. Their work aims to study if and how learners pick up the underlying rules in the data. In contrast, our setup allows investigating learners' preferences. In particular, we apply the description length metric to investigate learners' biases toward the mem and comp rules that explain the training examples in our setup.

**Sequence-to-sequence learners**    We use the following models when experimenting with SCAN:
**LSTM-s2s**: We chose the architecture used in Lake & Baroni (2017). In particular, Encoder and Decoder are two hidden layers with 200 units and embedding of dimension 32. This architecture gets 0.98 test accuracy on the i.i.d. (simple) split of SCAN, averaged over 10 seeds.
**CNN-s2s**: We use one of the successful CNN-s2s models of Dessì & Baroni (2019) that has 5 layers and embedding size of 128. We also vary kernel size in $\{3, 5, 8\}$ as it was found to impact the performance on SCAN (Dessì & Baroni, 2019). These architectures reach test accuracy above 0.98 on the i.i.d. split of SCAN, averaged over 10 seeds.
**Transformer**: We believe our work is the first study of Transformer's performance on SCAN. For both Encoder and Decoder, we use 8 attention heads, 4 layers, embedding size of 64, FFN layer dimension of 256. This architecture gets 0.94 test accuracy on the i.i.d. split of SCAN, averaged over 10 seeds.

**Training and evaluation**    We follow the same scenario described in the main paper with two differences: (a) when calculating $L$, we use blocks of size 1024, (b) during training, we sample $10^4$ batches with replacement, similar to (Lake & Baroni, 2017). We use batches of size 16 for LSTM-s2s & CNN-s2s, and 256 for Transformer. We repeat the training/evaluation of each learner 10 times, varying the random seed. Again, we use Adam optimizer and a learning rate of $10^{-3}$. Due to the large number of test examples and low performance of the

| | | FPA | | $L$, nats | |
|---|---|---|---|---|---|
| | *dropout* | count | mem | count | mem |
| LSTM-s2s no att. | 0.0 | 0.00 | 0.20 | 56.23 | **16.52**\* |
| | 0.2 | 0.95 | 0.00 | **0.17**\* | 60.15 |
| | 0.5 | 1.00 | 0.00 | **0.01**\* | 97.51 |
| LSTM-s2s att. | 0.0 | 0.32 | 0.68 | 63.30 | **47.68** |
| | 0.2 | 0.95 | 0.05 | **33.66**\* | 87.36 |
| | 0.0 | 0.99 | 0.00 | **7.84**\* | 121.48 |
| CNN-s2s | 0.0 | 0.00 | 0.55 | 1034.67 | **0.43**\* |
| | 0.2 | 0.00 | 0.98 | 999.62 | **0.01**\* |
| | 0.5 | 0.00 | 0.98 | 660.73 | **0.02**\* |
| Transformer | 0.0 | 0.00 | 0.65 | 261.02 | **1.17**\* |
| | 0.2 | 0.00 | 1.00 | 171.31 | **0.05**\* |
| | 0.5 | 0.00 | 0.97 | 116.34 | **11.10**\* |

(a) Count-or-Memorization

| | | FPA | | | $L$, nats | | |
|---|---|---|---|---|---|---|---|
| | *dropout* | add | mul | mem | add | mul | mem |
| LSTM-s2s no att. | 0.0 | 0.25 | 0.00 | 0.00 | **5.00**\* | 35.55 | 19.07 |
| | 0.2 | 0.30 | 0.45 | 0.00 | 12.07 | **11.04** | 37.39 |
| | 0.5 | 0.00 | 0.94 | 0.00 | 25.42 | **0.31**\* | 57.32 |
| LSTM-s2s att. | 0.0 | 0.00 | 0.11 | 0.58 | 25.09 | 36.33 | **12.09** |
| | 0.2 | 0.18 | 0.53 | 0.18 | 22.22 | **9.92**\* | 34.48 |
| | 0.5 | 0.00 | 0.98 | 0.00 | 30.26 | **1.40**\* | 58.84 |
| CNN-s2s | 0.0 | 0.00 | 0.00 | 0.65 | 236.41 | 247.52 | **0.42**\* |
| | 0.2 | 0.00 | 0.00 | 1.00 | 438.06 | 464.26 | **0.00**\* |
| | 0.5 | 0.00 | 0.00 | 1.00 | 318.12 | 346.19 | **0.00**\* |
| Transformer | 0.0 | 0.00 | 0.00 | 0.65 | 84.58 | 130.88 | **0.96**\* |
| | 0.2 | 0.00 | 0.00 | 1.00 | 65.62 | 99.05 | **0.02**\* |
| | 0.5 | 0.00 | 0.00 | 1.00 | 38.77 | 50.64 | **3.50**\* |

(b) Add-or-Multiply

| | | FPA | | $L$, nats | |
|---|---|---|---|---|---|
| | *dropout* | hierar | linear | hierar | linear |
| LSTM-s2s no att. | 0.0 | 0.00 | 0.00 | **16.38** | 17.72 |
| | 0.2 | 0.00 | 0.00 | **11.09**\* | 19.87 |
| | 0.5 | 0.05 | 0.00 | **7.76**\* | 15.46 |
| LSTM-s2s att. | 0.0 | 0.00 | 0.00 | 31.12 | **28.61** |
| | 0.2 | 0.00 | 0.00 | **10.63** | 17.55 |
| | 0.5 | 0.30 | 0.00 | **6.58**\* | 14.30 |
| CNN-s2s | 0.0 | 0.00 | 0.75 | 68.48 | **0.44**\* |
| | 0.2 | 0.00 | 1.00 | 99.42 | **1.16**\* |
| | 0.5 | 0.00 | 1.00 | 50.66 | **0.00**\* |
| Transformer | 0.0 | 0.05 | 0.00 | **3.99**\* | 8.56 |
| | 0.2 | 1.00 | 0.00 | **0.09**\* | 13.09 |
| | 0.5 | 0.69 | 0.00 | **1.21**\* | 8.76 |

(c) Hierarchical-or-Linear

| | | FPA | | $L$, nats | |
|---|---|---|---|---|---|
| | *dropout* | comp | mem | comp | mem |
| LSTM-s2s no att. | 0.0 | 0.00 | 0.00 | 32.66 | **11.15**\* |
| | 0.2 | 0.30 | 0.00 | **6.42** | 11.01 |
| | 0.5 | 0.00 | 0.00 | 42.65 | **38.55** |
| LSTM-s2s att. | 0.0 | 0.00 | 0.00 | 22.24 | **10.93**\* |
| | 0.2 | 0.45 | 0.00 | **4.38**\* | 35.14 |
| | 0.5 | 0.00 | 0.00 | **62.34**\* | 70.92 |
| CNN-s2s | 0.0 | 0.30 | 0.00 | **53.00**\* | 92.06 |
| | 0.2 | 0.75 | 0.00 | **0.37**\* | 110.50 |
| | 0.5 | 0.75 | 0.00 | **1.44**\* | 49.92 |
| Transformer | 0.0 | 0.00 | 0.05 | 129.65 | **27.12**\* |
| | 0.2 | 0.00 | 0.00 | 102.90 | **27.56**\* |
| | 0.5 | 0.00 | 0.00 | 147.83 | **6.36**\* |

(d) Composition-or-Memorization

Table 6: **Effect of dropout probability**: FPA measures the fraction of seeds that generalize according to a particular rule. Description length $L$ is averaged across examples and seeds. The lowest $L$ are in bold and * denotes stat. sig. difference in $L$ ($p < 10^{-2}$, paired t-test).

|  | Accuracy | | $L$, $\times 1000$ nats | |
|---|---|---|---|---|
|  | comp | mem | comp | mem |
| LSTM-s2s no att. | 0.00 | 0.00 | **8.3**$^*$ | 45.0 |
| LSTM-s2s att. | 0.00 | 0.00 | **9.1**$^*$ | 34.8 |
| CNN-s2s, kernel width 3 | 0.14 | 0.03 | **3.5**$^*$ | 19.0 |
| CNN-s2s, kernel width 5 | 0.26 | 0.02 | **2.0**$^*$ | 21.8 |
| CNN-s2s, kernel width 8 | 0.36 | 0.01 | **1.8**$^*$ | 24.3 |
| Transformer | 0.00 | 0.00 | **7.3**$^*$ | 23.7 |

Table 7: SCAN: Accuracy is averaged across seeds and examples. Description length $L$ is averaged across examples and seeds. The lowest $L$ are in bold and $*$ denotes stat. sig. difference in $L$ ($p < 10^{-3}$, paired t-test).

learners, all FPA scores would be equal to zero. Hence, we follow Lake & Baroni (2017) and use per-sequence accuracy, i.e. the ratio of the sequences with all output tokens predicted correctly.

**Results**  We report our results in Table 7. First, we see that CNN-s2s learners have a strong preference to comp, both in accuracy and description length. Furthermore, we observe that with an increase in the kernel size, description length of mem increases, while description length of comp decreases, indicating that the preference for comp over mem grows with the kernel width. We believe this echoes findings of (Dessì & Baroni, 2019).

While accuracy is well below 0.01 for all other learners/candidate combinations (and rounded to 0.00), according to the description length, Transformer and LSTM-based learners *also* have preference for comp over mem. This can be due to the transfer from compositional training examples, that can make comp explanation most "simple" given the dataset. Hence, the failure for systematic generalization in SCAN comes not from learners' preferences for mem. We believe this resonates with the initial qualitative analysis in (Lake & Baroni, 2017).

# E    STUDYING INDUCTIVE BIASES OF LM-LIKE ARCHITECTURES

While in this paper we focused on studying inductive biases of seq2seq architectures, we believe our entire framework can be used for investigating biases of architectures used in language modeling (LM). As language modeling emerges as a general few-shot learning mechanism where textual prompting is used as an interface (Brown et al., 2020), it becomes important to understand how models generalize in such a setup. In this Section, we demonstrate the applicability of our tasks and measures to such learners and scenarios.

We propose to study inductive biases of LM-like architectures by using prompt-based versions of our tasks akin to GPT2 (Radford et al., 2019) & GPT3 (Brown et al., 2020). As an example, we can train an LM-like architecture to continue the sequence "<sos>aaa<sep>" as "<sos>aaa<sep>bbb<eos>" and then prompt it with "<sos>aaaa<sep>" and check the output: is it "<sos>aaaa<sep>bbb<eos>" (mem) or "<sos>aaaa<sep>bbbb<eos>" (count)? Such formulation makes it trivial to adapt our tasks (Section 3) for the architectures used in language modeling. However, training differs from that of a language model due to the presence of the prompt.

In the following, we experiment with a variant of Transformer, namely joint source-target self-attention learner introduced by He et al. (2018). We use the code provided in (Fonollosa et al., 2019).[7] The joint source-target self-attention learner differs from the structure of Transformer in three ways. First, instead of attending to the last layer of the encoder in standard Transformers, here, each layer in the decoder attends to the corresponding layer in the encoder. Second, the separate encoder-decoder attention module and the decoder self-attention are merged into one attention, called mixed attention. The mixed attention is hence used to extract information both from the source sentence and the previous target tokens. Finally, joint source-target self-attention shares the parameters of attention and feed-forward layer between the encoder and decoder. The use of mixed attention and the sharing of parameters make the encoder/decoder distinction artificial. Indeed, we can see this learner as a "decoder-only" LM-like architecture that is adapted to performing seq2seq tasks in a way we described in the previous paragraph.

We use the same hyper-parameters of Transformer presented in the main text. Preliminary experiments showed that joint source-target self-attention learners do not succeed in the *Hierarchical-or-Linear* and *Composition-or-Memorization* tasks (i.e., less than 70% of the seeds learned the *training* set) when embedding size $S_{emb} = 16$ (the one used in the main experiments). We hence set $S_{emb} = 64$. The remaining hyper-parameters remain unchanged. Note that, according to Table 5, there is no switch of biases for any setting when considering

---

[7]Code link: https://github.com/jarfo/joint. However, unlike Fonollosa et al. (2019), we do not consider the locality constraint, and set the kernel_size_list parameter to None. This makes the architecture equivalent to the one introduced in (He et al., 2018).

|  | Count-or-Memorization | | Add-or-Multiplication | | | Hierarchical-or-Linear | | Composition-or-Memorization | |
|---|---|---|---|---|---|---|---|---|---|
|  | count | mem | add | mul | mem | hierar | linear | comp | mem |
| FPA | 0.00 | 1.00 | 0.00 | 0.00 | 1.00 | 1.00 | 0.00 | 0.10 | 0.00 |
| $L$, nats | 160.31 | **0.00**$^*$ | 71.55 | 121.09 | **0.00**$^*$ | **2.59**$^*$ | 74.90 | 96.52 | **57.94**$^*$ |

Table 8: Measuring biases of joint source-target self-attention learners: FPA measures the fraction of seeds that generalize according to a particular rule. Description length $L$ is averaged across examples and seeds. The lowest $L$ are in bold and $*$ denotes stat. sig. difference in $L$ ($p < 10^{-2}$, paired t-test).

$S_{emb} = 64$ compared to $S_{emb} = 16$. Specifically for Transformer, the only difference is that with $S_{emb} = 64$, the latter shows even a stronger bias towards the hierarchical rule in the *Hierarchical-or-Linear* task.

Finally, similar to Sections B and C, we only consider the *Count-or-Memorization* task with $l = 40$, the *Add-or-Multiply* task with $l = 20$, the *Hierarchical-or-Linear* task with $d = 4$ and the *Composition-or-Memorization* task with $M = 36$ and run 20 seeds per setting. Results are reported in Table 8.

First, when considering the arithmetic tasks (i.e., *Count-or-Memorization* and *Add-or-Multiplication*), we note a clear preference for the memorization rule (with a zero $L$ and 1.00 FPA) similar to Transformer behavior (see Table 5). Second, joint source-target self-attention learners are, similarly to Transformers, biased towards the `hierar` rule with 100% of the seeds generalized perfectly to the `hierar` rule and a significantly lower $L$. Finally, for the *Compositional-or-Memorization*, while only 1 seed generalized perfectly to the `comp` rule, $L$ shows a significant preference for the `mem` rule, in agreement with Transformers' biases.

Overall, our experiments in this Section indicate that the inductive biases of the "decoder-only" Transformer learner are very close to that of a standard Transformer seq2seq learner. On a higher level, we have demonstrated that our tasks and the description length measure can be applied for studying inductive biases of architectures that are often used in language modeling, indicating a high universality of the approach.

# F  EVOLUTION OF INDUCTIVE BIASES AT TRAINING

The description length measure that we defined by Eq. 1 represents both (a) how close the learner's generalization is to a fixed "explanation" of the data, and (b) if it can learn this explanation quickly, with few examples. In this Section, we aim to peak into the latter process, by looking how quick a learner can pick up a candidate explanation. To do that, we use the *Compositional-or-Memorization* task with 1000 different primitives.

We organize the experiment in the following way. We start by training learners on the training data, composed of all primitives and $M = 10$ (i.e., 10 compositional examples). We measure description length of the `comp` candidate generalization on the remaining (1000-10 compositional examples). Next, we train learners from scratch, but this time we add 10 additional compositional example (M + 10), and measure the description length on the remaining 1000-20 compositional examples. We repeat this procedure until all compositional examples are exhausted. As in this process the description length will decrease due to simply having less examples left for the evaluation, we normalize it by dividing by the number of remaining examples. We denote the result as $\bar{L}$.

Since this is a new task with a larger number of primitives, the hyperparameters used in main experiments wouldn't allow the models to reliably learn the training set. We hence use an embedding size of 64 for all architectures to have more robust convergence. For Transformer only, we also use a larger architecture with 2 layers. All the other hyperparameters remain unchanged compared to the experiments in the main text and we run 20 seeds for each setting. We report results in Figure 3.

Figure 3 shows that CNN-s2s are the fastest to prefer the compositional rule, starting already with low $\bar{L}$. Indeed, when provided by only 20 compositional examples, $\bar{L}$ is almost zero. On the other extreme, Transformers' behavior persists irrespective of $M$ with a large variance. While we note a decrease of $\bar{L}$ when increasing $M$, this tendency is very weak. Interestingly, while LSTM-s2s no att. learners start as less biased towards compositionality compared to Transformers, they display a lower $\bar{L}$ after seeing 120 compositional examples, and converge to an almost zero $\bar{L}$ with $M > 600$.

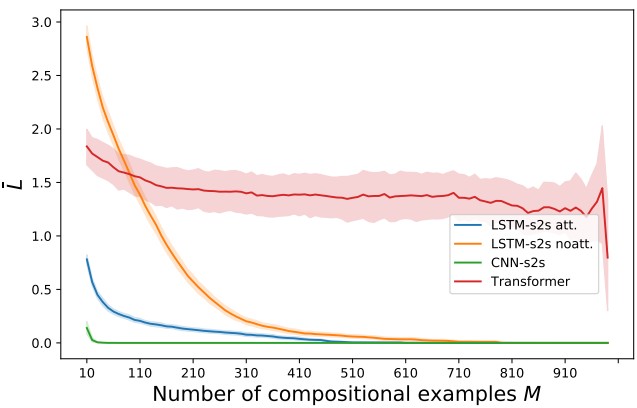

Figure 3: Per-example average description length $\bar{L}$ (nats), across 20 seeds for each learner as a function of number of compositional training examples. Shaded area represents 90% confidence interval.

