# OpenReview forum: "What they do when in doubt: a study of inductive biases in seq2seq learners"
_ICLR.cc/2021/Conference — ICLR 2021 Poster_

### Official Review · AnonReviewer2 · 2020-10-24
**interesting work**

**Rating:** 6
**Confidence:** 4

**Review:**

This work proposes to use description length (DL) to measure the seq2seq model's inductive bias.

Strength: It is clearly shown that DL gives more smooth measurements than FPA. Also, the experiments are principled, and well designed. And the conclusion are interesting and clear. I do believe that this framework can be utilized by future work in this direction to get a better understanding of seq2seq models.

Finally, the paper is well written, I am able to get the author's points.

Weakness: My major concern is the scale or completeness of the experiments. The authors concentrate on a training set of a few samples, which is far away from the usual large-data setting for LMs.

Moreover, the training data concerned only exhibit one or two kinds of bias, while in real data, there are usually various kinds of biases. I'm interested to see what the model will pick, when facing different kinds of structures in the data. For example, will the model favor easy rules than more complicated rules?

In addition to the different variants of seq2seq models, I'm also interested to see whether encoder-decoder model have different biases with pure LMs (only decoder, e.g., GPT-2).

---

> ### Author Response · Authors · 2020-11-17
> **Reply to AnonReviewer2**
>
> __In addition to the different variants of seq2seq models, I'm also interested to see whether encoder-decoder model have different biases with pure LMs (only decoder, e.g., GPT-2).__
> Thank you for this suggestion, that is indeed an amazing direction! We focused on seq2seq models as (a) they seemed to be of a higher application interest, (b) there is a lively area studying their inductive biases (eg SCAN by Lake and Baroni, 2018 and its follow-ups; (McCoy et al, 2020)) that we could connect to, and (c) operating with input/output pairs, as functions on sequences, is more naturally represented in the seq2seq framework.
>
> However, we believe that our setup can be directly applied to LM-like architectures via prompting. Indeed, we can train, for instance, a Transformer model to continue the prompt sequence "<sos>aaa<sep>" as "<sos>aaa<sep>bbb<eos>" and then prompt it with "<sos>aaaa<sep>" and check the output: is it "<sos>aaaa<sep>bbb<eos>" [mem] or "<sos>aaaa<sep>bbbb<eos> [count]? This diverges from the standard LM training due to the presence of the prompt during learning. At test-time, this approach does not differ from generating from an LM with a prompt (as in GPT-2 and GPT-3).
>
> Building upon your suggestion and using this idea to represent our tasks, we ran a set of experiments on a variant of Transformer (see Appendix E). We used the architecture described by "Layer-Wise Coordination between Encoder and Decoder for Neural Machine Translation", He & Tan et al., 2018. This architecture essentially merges Encoder into Decoder, while allowing different embeddings for inputs/outputs and resetting positional encodings at the start of the output sequence. Overall, we believe that this setup is extremely close to your suggestion of studying biases of LMs, while being compatible with our tasks.
>
> Our results (see Appendix E) suggest that this architecture has virtually the same inductive biases as its seq2seq counterpart: strong preference for hierarchical over linear reasoning, and preference for memorization over arithmetics and compositional generalization.
>
>
> __My major concern is the scale or completeness of the experiments. The authors concentrate on a training set of a few samples, which is far away from the usual large-data setting for LMs.__
> Our work aims to investigate learners' inductive biases. We place ourselves in the setting that maximally eliminates potential causes of success/failure that are orthogonal to the studied biases. While the tested setting differs from applications, it provides a very controlled framework to study learners' biases. Concretely, using this framework, our results constitute a more direct proof-of-concept for Elman et al. 1998's theory--an important theory in developmental psycholinguistic--and suggests that inductive biases can arise from constraints on the “wiring”.
>
> __Moreover, the training data concerned only exhibit one or two kinds of bias, while in real data, there are usually various kinds of biases. I'm interested to see what the model will pick, when facing different kinds of structures in the data. For example, will the model favor easy rules than more complicated rules?__
> Thank you for this interesting remark!
> We firstly want to notice that the notion of "easiness" is learner-dependent and can be closely connected to inductive biases of the learner. Indeed, we show the agreement between which rule the learners tend to follow when generalizing (that's what FPA tells us) with which rule they are faster  to learn  (="easier", that's what description length measures). Under those definitions our results confirm that indeed the learners favor rules that are "easier" to them!
>
> Secondly, we believe that our minimalistic tasks already offer multiple different explanations of the data  (c.f. the example provided in the introduction, which can be explained by an infinite number of rules). We choose to study alternatives that are intuitive to humans; experiments show that often learners generalize according to one of them. To humans, some of those would seem simpler than others (e.g. multiplication by 2 over, say, adding 10).
>
> Finally, we believe that in real-data datasets, there are enough cues to restrict the number of possible underlying explanations, probably in non-predictable ways. For example, to succeed in a machine translation task, learners need to encode semantic, syntax, and in some cases, morphology. In other words, real datasets will lower the size of the possible underlying rules, but they require learning different entangled components making it harder to connect learners' failure with their biases. For instance, if we consider the machine translation task again, a failure could be related to miss-learning semantic or syntax. This motivates our decision to focus on minimalistic artificial data.

---

> > ### Comment · AnonReviewer2 · 2020-11-24
> > **thank you for the response.**
> >
> > thank you for the response.

---

### Official Review · AnonReviewer1 · 2020-10-26
**Insightful analysis paper of seq2seq architecture biases**

**Rating:** 7
**Confidence:** 3

**Review:**

The paper studies inductive biases that are encoded in three main seq2seq architectures: LSTMs, CNNs, Transformers. Using one existing (fraction of perfect agreement) and one proposed metrics based on description length, and four dichotomy-like tasks, the authors show that, among other things, CNNs and Transformers are strongly biased to memorize training data, while LSTMs behave more nuancedly depending on input length and dropout. Unlike the first metric, the proposed metric takes values in a wider and more fine-grained range; the results are correlated for both of them. I also appreciate the attention to hyperparameter tuning and investigation of their effects in experiments. In general, the manuscript is well written and apart from a few minor questions can be accepted in its present form.

Questions:
- SGD was found to often generalize better than its adaptive variants, like Adam (e.g. Wilson et al. The marginal value of adaptive gradient methods in machine learning. In Advances in Neural Information Processing Systems. 2017), yet in your experiments there seems to be an opposite effect of changing the optimizer (Appendix C). Could you comment on why this is the case?
- Regarding the tendency of large dropouts to hurt memorization: to what extend does this help the peer bias in a task? It seems that hindering memorization seem to cause a complementary increase in count or add-mul ability (Table 6). Is there a value for dropout (or a combination with other hyperparameters) when Transformers would start showing a counting bias?

Minor:
- please use alphabetic literature sorting

UPDATE: I thank the authors for their detailed replies and running additional experiments. This resolves my questions and I'd keep my recommendation to accept the paper.

---

> ### Author Response · Authors · 2020-11-15
> **Reply to AnonReviewer1**
>
> __SGD was found to often generalize better than its adaptive variants, like Adam (e.g. Wilson et al. The marginal value of adaptive gradient methods in machine learning. In Advances in Neural Information Processing Systems. 2017), yet in your experiments there seems to be an opposite effect of changing the optimizer (Appendix C). Could you comment on why this is the case?__
> Thank you for pointing this out. In the main experiments, we investigated learners' biases when trained with Adam as it is the default optimizer for many applications. In particular, all state-of-the-art Transformer-based models in NLP are trained with Adam: BERT, ROBERTA, Transformer-XL, GPT3, etc.
> However, we have also explored the effect of SGD on learners' generalization in Appendix C. We found that it is hard to make all our diverse architectures learn the training examples when training with SGD. However, in the few instances where learners achieve a good performance at train time, we did not notice a difference between SGD and Adam results.
> Further, please note that our tasks do not have a "better" generalization, as all alternatives are equally valid given the training data. For instance, in the Count-or-Memorization task, memorizing the output could be seen as a perfect generalization as it explains the training set fully. Hence, preferring memorization when trained with SGD, as opposed to counting, does not contradict the advantage of SGD when testing generalization performance on less ambiguous tasks. In other words, our results do not show the opposite effect compared to Wilson et al. 2017 work; however, they do not show an advantage for SGD. This difference could be due to the fundamentally different setup. We clarified this point in the revised version of the supplementary material.
>
> __Regarding the tendency of large dropouts to hurt memorization: to what extend does this help the peer bias in a task? It seems that hindering memorization seem to cause a complementary increase in count or add-mul ability (Table 6). Is there a value for dropout (or a combination with other hyperparameters) when Transformers would start showing a counting bias?__
> Thank you for this suggestion. As you proposed, we ran 100 seeds of Transformers to solve the Count-or-Memorization task with dropout in {0.6, 0.8}. None of the instances learned the training example when dropout=0.8 (even when experimenting with lower learning rates). However, when dropout=0.6, we note 23% successful runs, of which 95% generalized perfectly to the mem rule (and 0% to the count one). Of course, we cannot be sure that there is no combination of hyperparameters where the Transformer will show a preference for count. However, we explored hyperparameter values that are coherent with what is used in the literature, and based on our extensive grid search, it seems unlikely to see Transformers preferring to count provided with only one example.
>
> __Please use alphabetic literature sorting__
> Thank you for your comment. We have updated the paper accordingly.

---

### Official Review · AnonReviewer3 · 2020-10-28
**Nice work on quantifying the inductive biases: Super interesting but no surprising and practical findings**

**Rating:** 7
**Confidence:** 4

**Review:**

The Topic of this paper, to investigate the inductive biases of different neural network architectures, is super interesting. They do this by considering the extreme case of when there is only one training example and define a set of biases based on the type of solutions the model converge to when all of the options are equally optimal based on the given training example.

The authors claim that minimum description length is a sensitive measure of inductive biases. As far as In understood the idea is that a solution (rule) can be simple for one learner, while more complicated for the other one, based on their architectural differences. And by measuring the minimum description length of data generate by a certain rule under a specific learner, we can tell how simple the rule is for the learner.  I am a bit puzzled by the way this claim is phrased (I don't have anything against the main argument though). Isn't minimum description length itself an inductive bias that is probably applied on all these models? The way I see this is that a model that achieves lower description length is finding a simpler solution and the simplicity of the solution can be characterised by the different solution preferences that is defined in this paper (e.g. memoization vs counting). I would really appreciate if the authors provide a clear distinction between different sources of the inductive biases and their interactions in their experiments.

A nice point about the paper is that in their experiments they evaluate the sensitivity of the preferences of different architectures to their hyper-parameters and it seems these preferences are consistent for most cases, but not surprisingly they observe that hyper-parameters like dropout rate can affects the generalisation behaviour of the models. That would be much nicer, if there were a bit more connections in the paper, about how these biases in the extreme case of only one training example cascade and affect the performance of the models when trained on larger scale datasets. For example, in a simple and still abstract case of gradually increasing the number of training examples, how the behaviour of these models change? Or in case of the point about few shot learning that is mentioned in the conclusion, I think it would be nice to have some experiments to show that the models will have these inductive biases and preferences even after being pre-trained on large datasets.

In the end I agree with the authors that
`Overall, our study shows that the inductive biases is more complicated than it seemed in these prior works and a more careful analysis is crucial.`

And while I think quantifying inductive biases is a super interesting, challenging and important problem to address, this paper needs a bit more work to be more impactful in terms of either providing us with practical insights about the inductive biases of different architectures, or providing us with benchmark or tools to be able to fairly compare the inductive biases of different models, or just by more clearly identifying the main challenges for doing this in a beneficial way (e.g., because of the complicated interactions between different sources of inductive biases and how this all changes when the amount of data and the capacity of the models increase/decrease).


***
Post Rebuttal: I like this paper and would vote for it to get accepted on the merits of: (1) Their finding about how MDL can be an indicator of inductive biases of the models; (2) introducing an experimental framework for studying inductive biases of the models. I'd also like to appreciate the authors' efforts to address reviewers concerns in the rebuttal. I agree with reviewer #5, that the paper can be better contextualised in the related research area but I think the paper is improved from this aspect a little bit during rebuttal and this is something that in general can potentially be fixed for the camera ready version.

---

> ### Author Response · Authors · 2020-11-18
> **Reply to AnonReviewer3**
>
> __Isn't minimum description length itself an inductive bias that is probably applied on all these models? The way I see this is that a model that achieves lower description length is finding a simpler solution and the simplicity of the solution can be characterised by the different solution preferences that is defined in this paper (e.g. memoization vs counting). I would really appreciate if the authors provide a clear distinction between different sources of the inductive biases and their interactions in their experiments.__
> Just to clarify, we never apply the minimum description length pressure on the learners, i.e. there is no external pressure for "simpler" solutions; the learners are trained in a standard way (optimizing NLL loss with a standard optimizer). Description length is only used as a metric to measure learners’ biases. We emphasized this point in the revised version of the paper (see Section 4.2).
> One of our interesting experimental findings is that the models' preferences according to the description length measure are in agreement with the preferences according to FPA. This shows that learners generalize according to the solutions that have the shortest description length (according to this learner). This implies that learners tend to "follow" prescriptions of Solomonoff's induction theory by "themselves".
> In Appendix, we investigated internal sources of biases, by varying different architectural and optimization hyperparameters. We found that the two main sources are (1) learners' architecture, and (2) dropout probability. Indeed, learners with different architectures (e.g., CNN-based vs. LSTM-based learners) show different and in some cases opposite biases. We also found that larger dropout values disfavor memorization preference.
>
> __And while I think quantifying inductive biases is a super interesting, challenging and important problem to address, this paper needs a bit more work to be more impactful in terms of either providing us with practical insights about the inductive biases of different architectures, or providing us with benchmark or tools to be able to fairly compare the inductive biases of different models, or just by more clearly identifying the main challenges for doing this in a beneficial way (e.g., because of the complicated interactions between different sources of inductive biases and how this all changes when the amount of data and the capacity of the models increase/decrease).__
> Thank you for this suggestion. We see our work as one that offers tools to compare the inductive biases of different models fairly. We show that description length is a robust metric to investigate learners' biases. It is a model- and task-agnostic metric that takes into account learners' size and their ease of learning. Crucially, we show that description length is not restricted to simple synthetic tasks as it is the case for FPA (or other intuitive accuracy-based measures that require some knowledge about the task). In Appendix D, we show how description length provides robust results when investigating learners’ biases using the SCAN dataset--a commonly used benchmark to look at systematic generalization. We integrated this point in the revised version of the paper (see section Discussion and Conclusion).
>
> __That would be much nicer, if there were a bit more connections in the paper, about how these biases in the extreme case of only one training example cascade and affect the performance of the models when trained on larger scale datasets. For example, in a simple and still abstract case of gradually increasing the number of training examples, how the behaviour of these models change? Or in case of the point about few shot learning that is mentioned in the conclusion, I think it would be nice to have some experiments to show that the models will have these inductive biases and preferences even after being pre-trained on large datasets.__
> Thank you for these suggestions. We investigated how description length L varies when gradually increasing the number of compositional examples M in the Composition-or-Memorization task. In particular, we look at the dynamic of L when learners are provided with M in {3,4…,39}. We describe the results in Appendix F.
> Figure 3 shows that CNN-s2s are the fastest to learn the compositional rule. Indeed, when provided by only 14 compositional examples, we get a description length approaching zero.  Interestingly, while CNN-s2s started less biased than LSTM-based models, the former displays a lower L for M>6. On the other extreme, Transformers’ behavior persists irrespective of M. When Transformers are fed with the whole dataset (M=40), they still have an L very close to the one with a sparse signal (M=3).

---

> > ### Comment · AnonReviewer3 · 2020-11-23
> > **Thanks for your response.**
> >
> > `>>>Just to clarify, we never apply the minimum description length pressure on the learners, i.e. there is no external pressure for "simpler" solutions; the learners are trained in a standard way`
> >
> > Thanks for the clarification. I find it indeed interesting that even without explicit bias for minimising minimum description length, the models are biased toward solutions with minimum description length. Another clarifying question, does that mean you don't have any regulariser such as weight decay? (besides your experiments with dropout).
> >
> > Moreover regarding your argument: "We also found that larger dropout values disfavor memorization preference." While, this is not a surprising finding I still think it is nice to empirically show that it holds, but you should also be careful in referring to the relevant literature and existing works that have already studied this, e.g.,: https://arxiv.org/pdf/1706.05394.pdf

---

> > > ### Author Response · Authors · 2020-11-23
> > > **Re: Thanks for your response**
> > >
> > > `I find it indeed interesting that even without explicit bias for minimising minimum description length, the models are biased toward solutions with minimum description length. `
> > > We really like this finding as it provides some "internal" consistency with Solomonoff's theory of induction. We are also intrigued if this could bring about new insights and connections between studying inductive biases and MDL?
> > >
> > > `Another clarifying question, does that mean you don't have any regulariser such as weight decay? (besides your experiments with dropout).`
> > > Thank you for this question! Indeed, we do not impose any type of regularization apart from the dropout (effect of which we study in Appendix). The training procedure is detailed in Section 4.2. Also, you can refer to the `hyperparams` folder in the anonymized zip archive for a detailed list of our training hyperparameters.
> > >
> > > `Moreover regarding your argument: "We also found that larger dropout values disfavor memorization preference." While, this is not a surprising finding I still think it is nice to empirically show that it holds, but you should also be careful in referring to the relevant literature and existing works that have already studied this, e.g.,: https://arxiv.org/pdf/1706.05394.pdf`
> > >
> > > Thank you so much! Indeed, it is relevant and it evaded our attention. We added the reference in the revised version of our paper.

---

> > ### Comment · AnonReviewer3 · 2020-11-23
> > **how description length L varies when gradually increasing the number of compositional examples**
> >
> > `>>>We investigated how description length L varies when gradually increasing the number of compositional examples M in the Composition-or-Memorization task. In particular, we look at the dynamic of L when learners are provided with M in {3,4…,39}.`
> >
> > What if M is way bigger (e.g. ~1e5, if it is at all possible to try that)?

---

> > > ### Author Response · Authors · 2020-11-25
> > > **What if M is way bigger**
> > >
> > > __What if M is way bigger (e.g. ~1e5, if it is at all possible to try that)?__
> > > We have re-run the experiment described in the new Appendix F but with 1e3 primitives and M (number of complex examples) varied from 10 to 1e3 with a block size of 10. We limited the experiment to 1e3 so that we can get results before the rebuttal period ends. However, we will investigate larger M values in our revision. We use the learners described in Appendix F.
> > > Next, since this is a new task with a larger number of primitives, the hyperparameters used in other experiments wouldn't allow Transformers to reliably learn the training set; and we didn't yet find better ones after a quick grid-search. Due to the rebuttal time limits we could not conduct a more extensive grid search.
> > >
> > > Results showing the remaining architectures are reported in this Figure, hosted anonymously ( https://drive.google.com/file/d/1II3UoTsOcv2xpDSJV3O87k9Q0GS84KsM/view?usp=sharing ). The figure shows that CNN-s2s learn compositional reasoning very fast. After a few compositional examples, they display almost a zero $\bar L$.  These learners are followed by LSTM-s2s att. which reach $\bar L \approx 0$ with $M \approx 500$. Finally, LSTM-s2s no att. require more than 600 compositional examples to have $\bar L$ close to zero. That is, our results show that learners, provided with enough evidence (by increasing M), would learn the underlying rule. However, the speed/sample-efficiency of learning is shaped by learners’ biases; the more biased a learner is towards the underlying rule, the faster it learns it. For example, LSTM-s2s no att., that  prefer the mem rule (see, e.g., Table 1 (d)) needed more than 600 compositional examples to reach $\bar L \approx 0$. We expect a similar but slower trend when experimenting with Transformers. We will add Transformers to the next revision after we find a hyper-parameters set that enables a reliable learning of this task.

---

> > > > ### Comment · AnonReviewer3 · 2020-11-25
> > > > **Thanks for the additional experiment.**
> > > >
> > > > Thank you very much for the additional experiments. I really appreciate your efforts in the rebuttal :)

---

> > ### Comment · AnonReviewer3 · 2020-11-23
> > **A related work**
> >
> > Your experiments on SCAN reminded me of this paper:
> > https://jair.org/index.php/jair/article/view/11674/26576
> >
> > The paper, compares the different seq2seq architectures, CNN-s2s, LSTM-s2s and Transformer-s2s, with regard to their ability to learn compositional rules (generalize compositionally).

---

> > > ### Author Response · Authors · 2020-11-23
> > > **Re: A related work**
> > >
> > > Thanks! We have added a discussion of this paper in the section with SCAN in the updated version of the paper.

---

### Official Review · AnonReviewer5 · 2020-11-07
**Why is this important?**

**Rating:** 4
**Confidence:** 3

**Review:**

The paper introduces a series of new datasets and task and investigates the inductive bias of seq2seq models. For each dataset, (at least) two hidden hypothesis could explain the data. The tasks investigated are count-vs-memorization, add-or-multiply, hierarchical-or-linear, composition-or-memorization. The datasets consists of one sample with varying length (amount of input/output pairs), which is denoted as description length. The models are evaluated on accuracy and a logloss. An LSTM, CNN, and Transformer are all trained on these datasets. Multiple seeds are used for significance testing. The results suggests that LSTM is better at counting when provided with a longer sequence, while the CNN and Transformer memorizes the data, but are better at handling hierarchical data. What this paper excels at is a thorough description of their experimental section and their approach to design datasets specifically for testing inductive bias, which I have not previously seen and must thus assume is a novel contribution.

However, I lean to reject this paper for the following reasons
- The paper tries to fit into the emerging field of formal language datasets for evaluating the capacity of deep learning methods. However, they do not build on any of the recent papers in the field. A new dataset, especially a synthetic one, should be well motivated by shortcomings of previous datasets and tasks in the field. I find the motivation and related works section lacking in that sense.
- We already know that LSTMs can count https://arxiv.org/abs/1906.03648 and that transformer cannot https://www.mitpressjournals.org/doi/full/10.1162/tacl_a_00306
- It is not clear to me why these results are important? Who will benefit from this analysis? Why are the current AnBnCn and DYCK languages that formal language people work with insufficient?
- LSTMs do not have the capacity to perform multiplication. I don’t know why your results suggest otherwise. You would need to incorporate special units that can handle multiplications in the LSTM, such as https://arxiv.org/abs/2001.05016

Update

First I'd like to thank the authors for their detailed rebuttal. I have upgraded my recommendation from 3 to 4. As mentioned in my review I believe this approach is interesting. However, as pointed by reviewer2, the experimental section lacks completeness. I think this experimental section would be suitable for a workshop, but not a conference. I am excited to hear you are considering to use this method as an inspiration for real problems. I'd like to see the paper resubmitted when you have obtained such results.

---

> ### Author Response · Authors · 2020-11-15
> **Reply to AnonReviewer5, part 1**
>
> Thank you for your feedback! Please find answers to your questions below.
> Before going into the details, we want to emphasize that our goal is not studying the capacity of neural models, as your summary suggests. Instead, we aim to reveal their inductive biases (systematic preferences in generalization). We illustrate this difference in the first reply. We believe that some of the points that you have raised, such as comparisons to the known model-capacity results and literature, could stem from this misunderstanding.
>
> __The paper tries to fit into the emerging field of formal language datasets for evaluating the capacity of deep learning methods__
> Please note that "evaluating the capacity" is very different from our goal in this work. We focus on finding inductive biases (systematic preferences in generalization). To highlight the difference using the "Add-or-Multiply" task: once our models learned to output a fixed sequence for a training example, they definitely have a capacity to memorize and output the same exact sequence for all new unseen inputs. However, our findings show that some of them "prefer" not to do so and chose other explanations of the training data (e.g., addition or multiplication).
>
> __However, they do not build on any of the recent papers in the field [of formal language datasets for evaluating the capacity of deep learning methods]. A new dataset, especially a synthetic one, should be well motivated by shortcomings of previous datasets and tasks in the field. I find the motivation and related works section lacking in that sense.__
> We connect to the ideas from works that are closer to our goal of measuring inductive biases (see the answer above): (McCoy et al. 2020) (in the NLP domain) and (Zhang et al, 2019) (in the image domain).
> McCoy et al. 2020 investigated learners’ biases towards hierarchical reasoning using still synthetic but less controlled datasets. The use of a more complex dataset that assumes the knowledge of different factors of language (e.g., semantics),  makes it harder to tie the failures of seq2seq learners with their biases. We differ by investigating seq2seq biases in a more focused setup, where we can disentangle between potential sources of failure. To this end, we introduce simple synthetic tasks. Moreover, in our setup, it is easy to look at other biases. Hence, unlike McCoy et al. 2020 work, we also look at arithmetic and compositional biases by introducing new tasks.
> Finally, our introduced framework is independent from the synthetic tasks. We show in the supplementary materials (see Appendix D) that our framework could be adopted when using existing datasets in the field, such as the SCAN dataset.
>
> __The datasets consists of one sample with varying length (amount of input/output pairs), which is denoted as description length. The models are evaluated on accuracy and a logloss.__
> We want to highlight that description length and lengths of examples are unrelated concepts. Description length is a principled measure that we propose to use to investigate learners’ biases. It can be applied to any input/learner combination independently of the length of the training example and can be even used in e.g. image-based tasks.
> For example, if we look at Table 1, we don’t see a systematic relation between description length and the length of the training example. Similarly, in the Composition-or-Memorization task, we investigate learners’ biases while varying the number of compositional examples and not their length.
>
> At the same time, description length is different from log-loss as well in e.g. that it catches the "speed of learning" effect, too (see "Calculating Description Length", p.2 in Section 2).
>
> __The results suggests that LSTM is better at counting when provided with a longer sequence, while the CNN and Transformer memorizes the data, but are better at handling hierarchical data.__
> Actually, our results show that CNN-s2s are the least biased towards hierarchical reasoning. Indeed, Table 1 ( c ) shows that 100% of CNN-s2s instances generalised perfectly to the linear rule, showing a clear preference for linear reasoning as opposed to the hierarchical one (see also Experiments section). Hence, when testing on hierarchical reasoning,  LSTM-based learners show a stronger bias compared to CNN-s2s. On the other hand, the latter architecture is better in handling compositional reasoning.

---

> > ### Author Response · Authors · 2020-11-15
> > **Reply to AnonReviewer5, part 2**
> >
> > __It is not clear to me why these results are important? Who will benefit from this analysis?__
> > One future work direction that we are very excited about is the transfer of these biases to real tasks. A concurrent related work is the one of Papadimitriou and Jurafsky, 2020 (https://arxiv.org/pdf/2004.14601.pdf). This work showed that frozen LSTMs could acquire the Spanish language if they were pre-trained on simple structured inputs. These structured inputs are as simple as learning hierarchical nesting. We find it exciting to see how these biases transfer that well when learning a real language. Our work could be very interesting in connection with such a study. It shows that some models, particularly Transformers, need as few as 4 pre-training examples to acquire the hierarchical reasoning. Also, Transformers with larger embeddings should perform better for transfer learning (see Table 5 ( c )).
> >
> > Besides, our findings can benefit researchers improving Transformer architecture (see, for example, Raffel et al. 2019 or Cordonnier et al. 2020). Indeed, equipped with this simple framework, we show that despite the architectural similarities between CNN-s2s and Transformers, the former performs better in compositional reasoning -- a property that is believed to be crucial for NLP tasks (see, for example, Lake et al. 2016). Thus, our results suggest that, by exploiting similarities between Transformer and CNN-s2s to guide the architecture search, one can build a hybrid architecture biased towards both the compositional and hierarchical reasoning.
> >
> > Finally, our work contributes to the long-standing nature-nurture debate in language acquisition. In particular, it constitutes a more direct proof-of-concept for Elman et al. 1998's theory -- an important theory in developmental psycholinguistics -- and suggests that inductive biases can arise from constraints on the "wiring".
> >
> >
> > __Why are the current AnBnCn and DYCK languages that formal language people work with insufficient?__
> > Note that the standard language recognition tasks, such as those used by (Hahn, 2020) are essentially binary classification problems. On the other hand, our study investigates the biases of seq2seq models. These models are typically used to map sequence of symbols to sequence of symbols with comparable input and output vocabulary sizes (as opposed to sequence of symbols to a binary output as it is the case for AnBnCn and Dyck languages).
> >
> > Hence, we go beyond of what was used before and experiment with mappings: A^n -> B^n (our "count-or-memorization" task), A^n -> B^{2n} ("add-or-multiply"), AnBAn -> B ("hierarchical-or-linear"). We note that the former two resemble AnBnCn languages, the latter is close to a Dyck language.
> >
> > At the same time, while having some resemblance to AnBnCn and Dyck languages, our tasks allow direct measurements of a diverse set of inductive biases of seq2seq models (e.g. for multiplication over addition) while preserving perfect ambiguity in the data and being closer to standard seq2seq use-cases.
> >
> > __LSTMs do not have the capacity to perform multiplication. I don’t know why your results suggest otherwise. You would need to incorporate special units that can handle multiplications in the LSTM, such as https://arxiv.org/abs/2001.05016__
> > Thank you for the reference! Please note that the paper (Madsen & Johansen, 2020), that you refer to, deals with multiplication of _pairs of arbitrary real numbers_. In contrast, we find that  LSTM-based seq2seq models can (and, in some cases, prefer to) multiply _small natural numbers by a fixed small natural-valued constant_ (2 or 3) when input and output are provided in the unary encoding.  We believe that these two setups are very different and require different representative power from the models. Hence, we believe there is no contradiction.
> >
> > We have also shared our code (anonymously) with detailed instructions for reproducing our findings. We included the multiplication results as the very first example. Hopefully this can help to resolve doubts w.r.t. whether LSTM-seq2seq indeed can do the limited-scope multiplication that we claim they can do.

---

> > ### Author Response · Authors · 2020-11-15
> > **Reply to AnonReviewer5, part 3**
> >
> > __We already know that LSTMs can count https://arxiv.org/abs/1906.03648 and that transformer cannot https://www.mitpressjournals.org/doi/full/10.1162/tacl_a_00306__
> >
> > Thank you for the pointer, and we added this reference in the Related Work section. We are aware of this line of work and discuss (Weiss et al. 2018) (that is close to your first reference) in Related Work. As we mentioned in our paper, we do not study learners’ ability to perform a given task, but their inductive biases (preferences).
> > The ability of LSTM to count does not necessarily imply that it will generalize to counting behavior when provided with other possible explanations. In our setup, we feed learners with training examples that are in agreement with multiple generalization rules. For example, in the Count-or-Memorization task, the training example could be described perfectly by either the count rule or the memorization rule. We know, based on prior works, that LSTM can generalize to both rules. What we found is that LSTM-based learners would more likely generalize to the count rule (>95 instances out of 100) provided by a long enough training example. We believe this finding is new.
> >
> > Similarly, we do not ask whether Transformers can perform counting on sequences of an arbitrary length. In the reference that you provided, the results are asymptotic and indicate that, as the input length increases, Transformers will start making mistakes. In contrast, we study their preferences (not abilities) on inputs, bounded in length.

---

### Decision · Program_Chairs · 2021-01-07
**Final Decision**

**Decision:**

Accept (Poster)

**Comment:**

This paper proposes a set of synthetic tasks to study and discover the inductive biases of seq2seq models.

Authors did a great job in convincing all the reviewers except R5 in their rebuttal. I do not find any serious concerns from R5's review. I personally think this is a very useful analysis paper.